# From “Information” to Configuration and Meaning: In Living Systems, the Structure Is the Function

**DOI:** 10.3390/ijms26157319

**Published:** 2025-07-29

**Authors:** Paolo Renati, Pierre Madl

**Affiliations:** 1World Water Community, NL-3029 Rotterdam, The Netherlands; 2Prototyping Unit, Edge-Institute, ER-System Mechatronics, A-5440 Golling, Austria; 3Department of Biosciences & Medical Biology, University of Salzburg, A-5020 Salzburg, Austria

**Keywords:** configuration, dissipation, meaning, symmetry breaking, order, coherence, stimulus–response, perception, qualities, relationship

## Abstract

In this position paper, we argue that the conventional understanding of ‘information’ (as generally conceived in science, in a digital fashion) is overly simplistic and not consistently applicable to living systems, which are open systems that cannot be reduced to any kind of ‘portion’ (building block) ascribed to the category of quantity. Instead, it is a matter of relationships and qualities in an indivisible analogical (and ontological) relationship between any presumed ‘software’ and ‘hardware’ (information/matter, psyche/soma). Furthermore, in biological systems, contrary to Shannon’s definition, which is well-suited to telecommunications and informatics, any kind of ‘information’ is the opposite of internal entropy, as it depends directly on order: it is associated with distinction and differentiation, rather than flattening and homogenisation. Moreover, the high degree of structural compartmentalisation of living matter prevents its energetics from being thermodynamically described by using a macroscopic, bulk state function. This requires the Second Principle of Thermodynamics to be redefined in order to make it applicable to living systems. For these reasons, any static, bit-related concept of ‘information’ is inadequate, as it fails to consider the system’s evolution, it being, in essence, the organized coupling to its own environment. From the perspective of quantum field theory (QFT), where many vacuum levels, symmetry breaking, dissipation, coherence and phase transitions can be described, a consistent picture emerges that portrays any living system as a relational process that exists as a flux of context-dependent meanings. This epistemological shift is also associated with a transition away from the ‘particle view’ (first quantisation) characteristic of quantum mechanics (QM) towards the ‘field view’ possible only in QFT (second quantisation). This crucial transition must take place in life sciences, particularly regarding the methodological approaches. Foremost because biological systems cannot be conceived as ‘objects’, but rather as non-confinable processes and relationships.

## 1. Introduction

Today, it is very common to hear about the key role of information in living systems as if this “thing”—i.e., information—were an ontologically “extra” element distinguishable from matter—a “magic entity”. This view has some fundamental problems in its conception of information, due to the ubiquitous, implicit postulate that reality can essentially be made up of “parts” that exist independently [1,2]. In a scientific culture that is completely aligned with the quantitative power of calculus and mathematics, this descriptive pre-assumption becomes inadequate when considering biological systems truthfully and realistically, i.e., avoiding mechanistic stereotyped representations of them. In fact, evidence within scientific contexts, points out that common sense is increasingly focused on “searching for the causes” of living dynamics by zooming in and resolving microscopic events, components and processes. However, this fails to recognize that what occurs at the microscopic level is a “how”, not a “why” [3]. Conversely, it could be said that we are typing these words on the laptop keyboard “because” some specific molecules are interacting, triggering efferent neurons to control selected muscles. If other molecules were interacting, we could be playing some chords on a guitar (!). However, this inconsistent ‘explanation’ for what an organism is displaying is ridiculous. Yet in “scientific medicine”, for example, it is implicitly accepted, confusing, again, “how” with “why”. Dissecting living systems into pieces to study their constituent parts disrupts the organismic continuum and the actual functioning that is possible only within a holonomic organisation. Taking them apart from their coupled environment disrupts crucial contextual relationships.

The aim of this manuscript is to provide the physical arguments that can consistently support the following points:(i)why adopting the category of “information”, as if it were a quantity, prevents us from achieving truthful knowledge of the dynamics of life;(ii)why, living systems can only be associated with ‘configurations’ (of their own boundaryless, open, living phase of matter and of the environment to which they thermodynamically and electrodynamically refer);(iii)how this analogical, qualitative, non-computational, and relational nature (and conception) of the living processes (not objects) and of “configurations”, is the only one capable of revealing the relevant aspect for the unfolding of life, such as meanings;(iv)how, despite being unable to reduce qualities to bits, algorithms, and models, the emerging picture of the dynamics of life—physically understood to be essentially analogical—provides us with incomparable epistemological richness and truthfulness.

For these reasons, this paper will not present any computations, models, data or experimental results, as the discussion needs to be conducted at a different level: the methodological and the conceptual one. According to the “view of the world” (*Weltanschauung*, in German) a corresponding idea of “rationality” emerges (to quote E. Husserl and M. Merleau-Ponty). Here, we aim to present those physical arguments that help to ground and build another “view of living systems”, which we believe is necessary to promote fruitful progress in science and overcoming the current hyper-technological era.

Following this initial introduction to the problem, the subsequent sections are organized as follows.

Section 2 reviews the traditional conception of information (à la Shannon) and explains why it is incompatible with living systems. It also demonstrates the ontological inconsistency between the presumed “software” and “hardware” and shows why the identity between information and entropy in living systems is untenable.

Section 3 reviews the physical basis for the emergence of coherence and symmetry breakings as well as the thermodynamic openness of the biological systems, focusing on (i) why the dynamical emergence of order, being lack of symmetry (only possible to describe in a QFT perspective), does imply the coincidence between structure and function (and between medium and message, as well as between hardware and software, psyche and soma, configuration and configured matter); and focusing on (ii) why the thermodynamic openness renders any approach that describes the features of living systems, due to an “information content” associated with a finite (computable, and arbitrary) number of degrees of freedom, meaningless.

In Section 4, we provide a more detailed description of how the oscillation modes ascribed to the nested hierarchy of coherences at work in the living phase of matter are responsible for (i) the biochemical pathways/physiology as a consequence of the coupling to the environment and (ii) compartmentalization, topological complexity (which prevents us from describing the organism as described by the second principle of thermodynamics, which applies to “bulk-like” matter only), and (iii) minimization of internal entropy through dissipation (which, again, makes inconsistent any deemed correspondence between “information content” and internal entropy). Moreover, this fourth section points out that coherence is an entropy-dissipating dynamics that allows the system to reach a lower fundamental (vacuum) level. This is a state constantly pursued by the organism (de facto, a time-crystal), representing its dynamical homeostasis, and enabling us to account for features such as teleology, adaptation and memory.

In Section 5, we demonstrate how the manifold of the unitarily inequivalent macrostates associated with the living system over time (which is materially and energetically open, but organizationally “closed”) implies a plurality of configurations that depend on its coupling with the environment, as well as on memory and on the most favorable possible configurations, which are non-locally explorable in the phase space (teleology). This condition enables us to understand (i) how the organism is, in its essence, a “relationship with the world” (and not merely “in relation” with it), (ii) that this relationship is of the “stimulus–response” (and not “action–reaction”) kind, (iii) that the configurations are not chosen randomly, nor pre-deterministically, and (iv) that merging memory and teleology yields the category of biological *sense*. This category is necessarily related to both to the past and the future and characterizes any biological state (and physiology) until the organism is sufficiently super-coherent, e.g., alive. This compels us to shift from a method that looks for the causes of a given behavior in living matter only via zooming into the microscopic scale, to a method that zooms out by looking at the biological sense and meaning of what happens in that relationship with the world. This is necessary as in doing so we are aware that what occurs at the microscopic level is orchestrated at larger scales by collective coherences. Such collective dynamics is the result of the coupling to the world and of the tendency to minimize the vacuum level of the coherent system. Moreover, we highlight in this section the physical difference between detection and perception, with the latter being ascribed to a change of the holo-state in a non-mechanistic nor random way. This implies the emergence of a meaning of the ‘reality configuration’ for the organism’s homeostasis, i.e., semantics, in a pre-linguistic, physical sense. Furthermore, we highlight how the idea of “qualia”, commonly adopted in neuroscience, is often used in the same way as “quanta”, which is, however, logically contradictory to the non-computable and analogical essence of the category of quality.

Section 6 provides some pertinent comments on the increasingly invasive use of AI as a tool to model complex systems, living matter, and even to mimic the human brain. The problems lie in the fact that all the databases and algorithms used to train AI platforms adhere to the “first quantization” paradigm, i.e., the conception of molecules, water (just seen as H_2_O molecule), and living matter based on QM and the idea of “information” as something reducible to bits or countable degrees of freedom. We address the practical consequences of this conception of “information”, showing how it fosters a collective mindset that will exacerbate the existing bias in future scientific literature by maintaining the “Meccano-type” idea of the living state, creating a false sense of the replaceability of living systems with machines and algorithms, thus further promoting a dangerous (and dystopic) transhumanistic ideology.

In Section 7, we offer some concluding insights, focusing particularly on a hopefully renewed methodology and sensitivity in life sciences.

## 2. The Inadequacy of Quantitative “Information” in Living Dynamics

The idea that we can retrieve the “raison d’être” for given complex emergent dynamics in living systems (such as physiology, morphogenesis, behavior, memory, etc.) by zooming into the microscopic level contains two inconsistencies. Firstly, we would have to take for granted the idea that molecular encounters occur on a local, short-range, and perhaps stochastic basis. Secondly, this would definitely prevent any form of teleology, memory, adaptation, homeostasis, behavior or autopoiesis (all of which are typical features of living beings), all apparent even in the most archaic forms of life (such as bacteria, protists, etc.).

In the following sections, we will briefly summarize the physical basis of non-randomness (i.e., minimal entropy-keeping), by acknowledging the water-based (super)coherent nature of biological matter and reconsidering how “information” in the living state should be understood and reconceived. This will provide important insights into the emergence of semantics in living systems, i.e., how living beings are susceptible to and generators of meanings, and what “meanings” actually are, physically.

As we will see, semantics is often associated with the topic of “information” as defined by C. Shannon, which works very well in telecommunications. However, in biological contexts, meaning does not emerge intrinsically from what is called “information content” in IT. As Shannon himself warned [4], this concept of information should not be considered applicable in general, especially not in biology. In living systems such a technical concept should not be associated with the idea of a “signal” being transmitted from somewhere to elsewhere, nor should it be bound to the idea of a discrete sequence of “units” (bits, q-bits, digits) [1,5].

-It is assumed that information is something that is formed “on/above” matter, like a “dress”. Whereas, if we think rigorously, there can be no “form” without “something to be formed” which, in turn, if “partial” is, again, a “form”. When we consider the fundamental quantum (fluctuating) nature of any matter/energy system, as we will discuss below, no “part” can exist by itself.-The ‘quantity of information’ is associated only with ‘individual’ or ‘fixed’ degrees of freedom for ‘components’ (bit-carriers), and not with their relations or interactive dynamics. The fact that a portion/a component of the system, considered as a bit, is placed together with certain others, or enters a given process before instead of after (let us think of nucleotides or ribosomes, for example), constitutes “information”, not so much by the fact that such a component/bit-carrier is present or absent, but because of its context (corresponding, moreover, to the pragmatic aspects in linguistics).-The implicit idea that information (seen as a set of bits) is something made of “quantities”, “parts” or “series of quanta/bits”, goes hand in hand with the assumption that such information can be ‘innocently’ separable from the medium/channel, and that the physical reality can be split into software and hardware. In fact, no “hardware” remains the same, regardless of the type of bits it embeds: storing a file/software on a hard drive, for example, involves arranging electric charges or spins ‘here and there’ in space, instead of other configurations. Software is nothing but how the hardware is arranged, and the latter is the consequence of the configurations of the same (or lower) degrees of freedom.-In Shannon’s definition of information, which is suitable for telecommunications and informatics, the amount of information increases as the uncertainty in the emission of the source increases. This is why it correlates with the entropy/fuzziness/disorder of the transmission apparatus [4].-In these ambits, information is not considered in relation to its context. A bit-sequence carries the same “message” regardless of the place/time/medium; it is defined as an invariant.-For this final reason, information in Shannon’s sense has nothing to do with meaning. A set of bits, or any element/part of reality, has no inherent meaning, since this category—crucial in living systems—can only emerge out of relationships.

However, unless we trickily resort to metaphysical additions or ontological discontinuities to “explain” the non-trivial analogical features of biological systems (such as teleology, adaptation, behavior, semantics, and memory, even ‘consciousness’, whatever that may mean), we must recognize that they are also matter systems. Therefore, a conception of “in-formation” that is not physically consistent with the dynamics that underlie their emergence is neither fundamental nor adequate to be implicated in living processes [1,5].

In living organisms, “information” cannot be understood as a “software”, superimposed on their matter/structure (“hardware”), but must be seen as the network of functional, topological, and semantic relationships that actually exist within their materiality. The coherent water connectome, ions, field gradients, quantization of oscillation modes, DNA, RNA, proteins, and metabolic cycles are both the medium and the content, the message and the messenger. Regarding biological information and meaning, McLuhan had already stated that “the medium is the message” because it (the medium) is literally the content (the message), and the content of the message is unique to that medium [6]. This ‘message’ can only be that ‘medium’ (i.e., for example, a set of molecules with a given shape, electrical charge, oscillation patterns, specific biochemical reaction partners, at a given reaction step, at a specific reaction site, etc.). It cannot be “separated” from such an irreducible network of relationships (the former is ‘forged’ by and written in the latter). The proper configuration of the system/medium (in space and time) determines the so-called “information”, that is, the suitable number and types of degrees of freedom and work cycles needed to maintain far-from-equilibrium states along with their internal system organization (i.e., compartmentation, differentiation, and potential gradients).

If we understand that order emerges dynamically, and that a living organism is a thermodynamically open system—a process, not an object—consisting essentially of its coupling with its environment (its thermodynamic double [7]), then the concept of “information” gives way to a relational one: the dynamical configuration (in space and time) which is not fully mathematizable. The naïve, clumsy “information content” (an erroneous expression, as it is still based on the “quantity” assumption) increases as the constraints on, and the relations between, the degrees of freedom become more differentiated [1,8,9].

In living systems, “information” is ultimately the opposite of entropy (as understood in computer science or telecommunications). The greater the constraints (temporal and spatial order requirements) on kinds, forms, places, paths, moments and frequencies of the energy and matter exchange, the greater the configuration complexity (misnamed “informational content”).

## 3. Dynamic Emergence of Order, Symmetry Breakings, and Dissipation

In the living realm, there is an irremovable isomorphism between the medium and the message, despite them still being mistakenly referred to as “two” separate entities due to limitations in our language. This fundamental inseparability is reflected in the transcendence of a Cartesian dualism between psyche and soma—that is, between function and structure, which are two sides of the same coin and cannot exist independently of one another. In the dualistic view, it is as if a living organism were a system of matter “enriched” with “information” (thought of as something that comes and goes independently). This is not so different from a metaphysical view, in which the relationship by which something non-physical (psyche) influences something physical (soma) determines its functioning and expression—a perspective that would remain entirely vague [2].

A truly quantum description of condensed matter (2nd quantization) takes into account the multiple levels of vacuum states where virtual excitations become stably coupled with matter quanta—such as electrical charges, atoms, and molecules depending on the scale in which we are interested—triggering their collective oscillation. It is at this point that the particle-wave duality dissolves, allowing a unified picture of the world to spontaneously emerge and enabling us to understand the true dynamical origin of order in biological (and condensed) matter. This unified picture makes it possible to develop proper conceptions of the “mysterious” logos underlying the living dynamics (free from any vitalistic naïvety) [2,7,10].

For decades, biological matter has been attributed to long-range correlated components [11]. However, this property is also characteristic of inanimate condensed matter when viewed through a ‘field perspective’ [12]. Yet these correlations behave radically differently therein. Therefore, if we presume that the components constituting the various phases/states of matter (including the inanimate and living phases) are always exactly the same, we might be tempted to think that something “magical” (added up) causes the components in living systems to behave in a “special way” so as to yield a living organism. This vitalistic view remains the only one possible if we adhere to a classical or a semi-classical view of matter ignoring the progress made in physics. In the conceptual framework of QM, a collection of interacting molecules implies that those molecules remain always the same. This is in accordance with von Neumann’s fundamental theorem [13], which allows one unique ground state (vacuum). Accordingly, only one unique state (or phase) is permitted by this theorem, and phase transitions cannot be described [13]. On the other hand, a classical mean-field approach, as used in dynamic systems, ref.[14] is often insufficient to account for some complex features, such as those occurring in living organisms (listed previously). For example, classical statistical mechanics did not lead to a substantial understanding of neural cooperation phenomena in human brain [15], as the electric and magnetic fields associated with neuronal currents are too weak to be responsible for the observed massive neural correlation. Furthermore, as has been masterfully demonstrated, the spread of chemical agents and neurotransmitters is too slow to explain the observed global and synchronized activity [16,17,18,19]. Similar unresolved questions concern the understanding of even simpler components and processes in living matter, particularly the existence of chemical cycles (see Figure 1) involving biomolecules, biological signaling, and biocommunication within and outside of cells, biological clocks and synchronization, hormonal secretions, spatial organization in morphogenesis, and nested topology in organisms [20,21]. The mechanism of spontaneous symmetry breaking (SSB) is a key tool for understanding not only “particle” physics and condensed matter [22,23] but turns out to serve as a “gold standard” also for a proper understanding of biological systems [24,25]—including brain states too [17,18,26]. It enables not only the description of systems exhibiting emergent ordered patterns (spatial order), as observed in crystals and magnets, but also of time-ordered dynamics (motional order), which are clearly visible in superfluids and superconductors [12].

Biological systems exhibit order in both space and time, yet they generate a type of order that is not repetitive (as E. Schrödinger had previously observed [28]). Thanks to dissipation, this order can be configured in an “infinite” number of ways due to its coupling with the environment [26]. A biological system essentially consists of, or exists as, a specific ‘coupling to the environment’ (we underline this point: it does not exist as “an entity that is coupled”, rather, it exists essentially as “coupling, response, and process”). To maintain coherence, it must maximize the uncertainty in its number of quanta, N (ΔN→∞), which is a quantity related to the phase of oscillation, φ, by the uncertainty relation (Figure 2) (in natural units) [12]ΔN·Δφ ≥ ½(1)

Coherence is a collective state in which the phase is maximally defined and shared by all co-resonant oscillators (Δφ→0). It is established via thermodynamic openness, i.e., by expelling entropy into the external environment, and it becomes further stabilized by continuously exchanging quanta (matter and fields) with the environment, maximizing the uncertainty of the number operator [22,32]. Living beings exist as relations and flows, not as closed, finite objects/bodies [7,33]. This is another reason why the quantitative, bit-like concept of information is completely inapplicable. As a living being is intrinsically an open system, any attempt to isolate it eliminates its functionality and results in destruction (death).

As Vitiello has well pointed out [7,10], the application of thermodynamics (adequate for isolated or closed systems) requires that, when studying an open system, “*α*”, we must “close” it by taking into account the environment in which it is immersed, “β”. In doing so, the flows of matter, energy, etc., between the system (*α*) and the environment (β) are constantly balanced, enabling us to describe the set (*α*, *β*) as a “closed” system, for which there is no energy flow. In a closed system, energy is conserved, thus the relation E(*α*) − E(*β*) = 0 must hold for any state of the set (*α*, *β*). This energy balance E(*α*) − E(*β*) = 0 is equivalent to the relation N(*α*) − N(*β*) = 0, where E(*α*) and E(*β*) denote the energies associated with the number of quanta involved in the exchange. N(*α*) and N(*β*) of *α* and *β*, respectively, are the quanta condensed in the fundamental vacuum state of (*α*, *β*), i.e., E (*α*) = Ω N(*α*) and E(*β*) = Ω N(*β*), where Ω is the energy of a single quantum. This relationship, which describes the energy balance between *α* and *β*, is full of meaning. For example, it tells us that N(*α*) and N(*β*) can certainly vary, provided that these variations cancel each other out. The fundamental state of the entire system (*α*, *β*) must be a condensate of an equal number of quanta attributable to the sub-systems *α* and *β*, meaning that N(*α*) = N(*β*) holds at all times in the history of system *α*. This becomes even more meaningful once we consider that ‘order is symmetry breaking’ and has a ‘dynamic origin’ [10].

In QFT, it can be shown [22,23] that the existence of long-range correlations and order, as well as their associated quanta, the so-called Nambu–Goldstone (NG) bosons (or their corresponding bosons in finite systems, acting as phasons), are indeed a consequence of the breakdown of the symmetry for given degrees of freedom. Consider, for example, a collection of atoms in a state where their positions can be shifted without producing observable changes in the macroscopic system (i.e., there is symmetry under spatial translations; this is the case for a gas, or a liquid, or an amorphous system, which are highly disordered). When such a space-translational symmetry is broken, meaning that the atoms can only occupy specific positions at defined distances (multiples of the lattice step), the atoms cannot occupy the same equivalent variety of positions as before: they are forced to occupy lattice positions unless they spend energy [10]. This is the case of a crystal.

However, the most interesting aspect is in *how* this happens, and that it is not due to short-range forces, as is usually assumed in the generally accepted view of condensed matter. As the late Prof. Giuliano Preparata demonstrated, [12] if the atoms in a gas are consistently considered quantum objects, they can do nothing but fluctuate (obeying Heisenberg’s principle). As they are made of electric charges, they are inextricably linked to their coupled radiative electromagnetic fields, which propagate in space over distances two to four orders of magnitude larger than the size of the atoms themselves [12].

A general principle has been demonstrated in QFT with liquid water being the most meaningful case [34]: an ensemble of a large number N of quantum particles (e.g., electric charges, dipoles or multipoles), is always bound to oscillate as quantum objects when coupled to a background radiative em-field, above a critical volume density and below a temperature threshold. In this state, the ensemble displays a spontaneous transition from the original non-coherent state to a coherent one. In doing so, the ensemble exhibits a well-defined oscillatory phase and attains a lower ground energy level than the original non-coherent ensemble. This difference in energy is termed ‘energy gap’. The existence of the energy gap ensures the stability of the coherent state against external perturbations and prevents the individual components from transforming independently of the ensemble [22]. The system is now defined by quasi-particles, which are the equivalents of Nambu-Goldstone bosons. These quasi-particles arise from the coupling between matter and the em-field (and its self-trapping) and correlate the matter components over large distances (roughly equal to the wavelength of the electromagnetic mode at which the matter components resonate) [12]. The appearance of such bosons (acting as phasons) at the new, lower, ground level, i.e., the new vacuum state, is the condensation of the quanta that define the macroscopic wave function of the system (the crystal in this example). These larger-scale ensembles of quanta now exhibit a classical property, in that they are no longer subjected to quantum phase fluctuations and can, therefore, be treated as a classical field: the order parameter [10,22]. The lowering of the vacuum level, with the appearance of an energy gap, which expresses the amount of energy required to push a matter component out of the coherent state, is related to the thermodynamic openness of the system. In other words, the system must be able to dissipate an amount of energy equal to the energy gap per atom in the form of heat (the latent heat of the condensation, expelled as entropy) in order to settle in the new, thermodynamically favorable, coherent state (see Figure 3 for the case of liquid water) [34].

Thus, in this simple example, the crystal order of the atoms in their lattice sites emerges as a dynamical effect of symmetry breaking; said differently, crystal order is the absence (or reduction) of spatial translational symmetry. In QFT, the dynamics that governs the behavior of the elementary components of a physical system in order to generate the ordered structures has general properties that can be found in the physics of elementary particles, in condensed matter, as well as in cosmology and in biological systems [23]. Ergo, in the broadest sense, [36], order is lack of symmetry.

In the above example, the NG-like bosons correspond to the collective lattice modes, i.e., phonons. These bosons exist in the system as fully characterized excitations (quanta), just like the elementary components (atoms) whose long-range correlations they are responsible for. Scattering techniques have been used to observe them by using them as targets of particles, such as neutrons, used as probing excitations [37]. Studying the energetic spectrum by exciting them, inducing deformations in the ordered structure (e.g., the crystal lattice), and submitting it to external tensions or thermal jumps reveals variations in their density of states. These correlation quanta then enter the list of elementary components of the system as structural elements, i.e., phonons are the correlation mediators, they are structural. They are an integral part of the structure, are real and proper elementary components of the system.

However, they cannot be “extracted” from the system in the same way that an atom or a group of atoms can be extracted from their lattice sites in a crystal. The NG quanta are only associated with the state of the crystal. There are no phonons that can propagate freely out of a crystal; they only exist as long as the crystal exists. Above the crystal’s melting point, only the atoms that constitute the crystal prior to fusion are detectable, with the phonons no longer be present. The latter “constitute” the collective state within the network of atoms in the lattice function—they are, in fact, the collective way of being of the atoms in the crystal-state function. Therefore, the NG quanta are identified with the specific order and the structure–function for which they are responsible. Without them, the system “is another system” with completely different physical ordering and properties.

In this vision, structure and function are inextricably linked (entangled) and co-substantiated [10]. In condensed matter physics, it is impossible to make a meaningful distinction between structure and function. Thus, the intrinsically dynamic vision of QFT offers a unified view of the condensed system that is no longer divided in structure and function as occurs in the static (classical or corpuscular QM) view. The latter is based on the ontological prejudice (see references [1,10]) of the pre-existence of an isolated, self-contained structure, disconnected from any functional reference. This is crucial for understanding how the living phase of matter can emerge and for distinguishing consistent descriptive approaches from inappropriate ones, especially with regard to thermodynamics, as will be discussed in the following section.

In reference [29], we showed that the quantum foundations of the emerging living phase of matter cannot be framed within a “quantum biology” based on QM. Reference [8] provides a review on how the living phase of matter is based on nested coherences of the water-based connectome (which constitutes more than 98% in molar fraction for an average human adult). Living matter is an interaction of a water connectome with suitable biochemical species, resulting in ordered patterns in both time and space—not only at a microscopic scale, but at all scales involved in this hierarchy of coherences. The ordering of biochemical reactions over time, along with all recognizable work-cycles (physiology), constitutes the “law of existence” of the organisms and evolves in a non-random manner. However, as they essentially constitute the coupling with the environment and the process of irreducible response to stimuli, they cannot be pre-determinable a priori [33].

Ordering in motion implies the persistence of a well-defined (multiplexed) phase throughout the oscillations of the elementary components. Ordered patterns cannot be derived as the straightforward sum of the properties of the components. For example, even in inanimate matter, macroscopic properties such as magnetization, electrical properties, stiffness, etc., are system properties that regard the whole, and not of the individual atomic or molecular components they are made of.

As we have seen, any kind of order implies that symmetry with respect to time (before/after) or space (translations, dilations/contractions and rotations, etc.) transformations is broken: here and now is not equal to there and then. Therefore, when degrees of order appear, some symmetry and invariances in spacetime are broken [25]. Hence, the time-history of the states undertaken by a living system, α, can be thermodynamically described through the formation of each configuration (*α*, *β*) that occurs as a consequence of symmetry breaking of induced via the coupling with the environment, β. Thus, the validity of the relation N(*α*) − N(*β*) = 0 for any state implies that:(i)the multiplicity of the possible configurations is permitted by the existence of the multiple (infinite) possible fundamental states within the QFT scenario;(ii)the coexistence of these multiple configurations is due to the fact that the fundamental states are orthogonal to each other [10,22];(iii)their succession in time is given by the dissipative dynamics, i.e., their thermodynamic history, which is defined as all the possible couples of values of N(*α*) and N(*β*) that obey the relationship N(*α*) − N(*β*) = 0.

All of these potential pairings correspond to possible “organism–environment” configurations, which have qualitative implications for the coherence of the former (i.e., meaning, physically speaking, not linguistically). This shows why it is paramount to pay attention to semantics and meaning when dealing with living dynamics and demonstrates also that the succession of system’s states is in fact a time-dependent thermodynamic (dissipative) history. Accordingly, a living system depends on (i) the previous states in a deterministic but a priori unpredictable way, and (ii) possible states that have not yet been manifested and are non-locally explored by quantum fluctuation in the system’s phase space (which explains teleology) [8,9].

## 4. Coherence, Cycles, Compartmentation, and Entropy Dissipation

Self-trapped fields can shape the topology of a system as stationary waves wherever coherence domains are present, because field gradients across the boundaries of coherence domains imply the development of ponderomotive and dielectrophoretic forces. These forces, together with the interference patterns, create anisotropic distributions of matter in space and time. Thus, the coherence, established on the aqueous connectome (which constitutes about 99% of any living system [38]), not only orders biochemical activity within time-sequenced, interlocking cycles but also transduces into a vast hierarchy of compartmentation at nano- and micro-scales. These nested coherences extend from molecular scales to encompass vesicles, cells, tissues, and organs, eventually reaching the whole organism [20,39]. Evidence has indeed accumulated that the morphogenesis of structures in living matter could result from the spatial interference patterns of standing electromagnetic waves in aqueous media [40,41,42]. This phenomenon, known as cymatics, is observed in mechanical and acoustic waves (see Figure 4). [43,44] Cymatics essentially consists of studying how matter is patterned in space by oscillating acoustic fields [45].

Cymatic and dielectrophoretic dynamics result in the breaking of the homogeneity of the distribution of matter in space. A system in which standing waves are present becomes ultra-structured, consisting of zones where matter components are concentrated (nodes) and zones where they are rarefied (antinodes). This occurs on a scale comparable to the wavelengths involved. In three dimensions (3D), we observe patterns of matter forming membranes, chains, stacks, vesicles, and so on. These distributions of matter occur because the “moulds” created by the interference pattern are shaped by the multiplicity of standing waves as well as by the ponderomotive forces generated by field gradients [42]. For a spherical particle with a given radius *r* and dielectric permittivity *ε_p_*, immersed in a homogeneous medium with a dielectric permittivity *ε_m_*, in the presence of a time-averaged electric field gradient ∇E_rms_ (“rms” being “root mean squared”) a ponderomotive (or dielectrophoretic) force acts in the following way:(2)$$F_{DEP}\geq 2\pi r^{3}Re\left\{\frac{\varepsilon_{p}^{*}-\varepsilon_{m}^{*}}{{\varepsilon_{p}^{*}+2\varepsilon }_{m}^{*}}\right\}\nabla {\left|\vec{E_{RMS}}\right|}^{2}\;$$

The self-trapping of the electromagnetic potential ***A*** inside the CDs generates strong gradients at the interface that are associated with the generation of ponderomotive forces [38,47]. According to the laws of electrodynamics, these forces develop into two contributions: one electrostatic (independent of the frequency) (Equation (3)) and one electro-dynamic (dependent on the frequency) (Equations (4) and (5)). Let *C* be a constant; then, the following is true:(3)$$F_{rep}=-\frac{Q^{2}}{M}\nabla {\left(A\right)}^{2}\approx qV\;(V=0.1\;Volt)$$(4)$$F_{i-CD}=C\frac{\left(\omega_{CD}^{2}-\omega_{i}^{2}\right)}{{\left(\omega_{CD}^{2}-\omega_{i}^{2}\right)}^{2}+\Gamma^{2}}\nabla (A^{2})$$(5)$$F_{int1-2}=C\frac{\left(\omega_{1}^{2}-\omega_{2}^{2}\right)}{[{\left(\omega_{1}^{2}-\omega_{CD}^{2}\right)}^{2}-{\left(\omega_{2}^{2}-\omega_{CD}^{2}\right)}^{2}]+\Gamma^{2}}\nabla (A^{2})$$

Equation (4) shows that any electric charge on the surface of a CD is repelled outward with a force proportional to the ratio *q*^2^/*m*, regardless of its sign. As the static force is always repulsive for both positive and negative charges, it depends on the charge-to-mass ratio. Electrons, 2000 times lighter than protons, are repelled much more strongly than the heavier nuclei. The molecules then become strongly polarized and therefore chemically more reactive [38]. Equation (4) describes how the interaction force between a CD and an atomic or molecular species depends on the vibrational modes of both. In other words, if a species has (at least) one oscillation mode whose frequency is very close to that frequency of the em-field trapped in the CD and decaying outward, it is subjected to a diverging force (typically attractive) that increases as the frequency difference disappears. Equation (5) describes the attractive/repulsive force between two species according to the same dynamics, as a function of their frequency matching that of the background field of the CD oscillating at *ω_CD_* [3,38].

The standing waves at work involve multiple coherence levels. Indeed, coherent oscillations are maintained over time and are confined within the boundaries of the coherence domain, extending over spatial ranges of the same order of magnitude as the wavelength of the electromagnetic mode coupled to the oscillating matter components. Their timescales are of the order of magnitude of the inverse of their frequency [22,25,48,49]. This implies the formation of compartments and matter structures whose topology is dictated by the shape of the electromagnetic potential gradients and forces [8]). As briefly mentioned above, 2D (membranes and sheets) and 1D (chains and filaments) structures have been shown to be the result of field gradients at coherent interfaces and of electric and magnetic fields self-focusing [24,25] associated with the Anderson-Higgs-Kibble mechanism [50,51,52,53]. One of the typical characteristics of living systems is their essential functioning based on cycles and the fact that their activities are predominantly rhythmic.

Many efforts have been made in vain to identify a ‘control centre’, and more recently, to identify “master genes” that govern biological rhythms [54]. In fact, every organism, even the simplest, is dictated by and characterized through a vast set of cycles with periods ranging from pico- and nanoseconds (for collective electron oscillations), to micro- and milliseconds (for the work cycles of larger “molecular machines”) [55]. Such cyclic activity performs bio(electro)chemical work spanning huge timescales, from electron motions in redox reactions, up to steric changes of enzymes and proteins, membrane pulsations in cells. This extends to circadian and seasonal cycles of organisms, populations and even across ecosystems [48,56].

To gain a deeper understanding of the vast topic of thermodynamics in living systems, it is helpful to direct the reader to the works of Onsager [57], Morowitz [58], Ho and Ulanowicz [49,59], Prigogine and Stengers [60], and Jørgensen and Svirezhev [61]. For the purposes of this paper, it is useful to recap and discuss some key issues relating to cycles and their role in energy mobilization. Dissipation is one of the most important aspects of the functioning of biological matter as it is closely related to energy flow [62]. However, the energy flow is not quite the main point. Consider, for instance, one of the simplest and most prototypical dissipative structures: the Bénard–Raleigh cells (see Figure 5), which form in water (or a more viscous liquid, such as paraffin) inside a flat pan heated uniformly from below [63]. At a critical threshold temperature, when the difference in temperature between the cooler upper surface and the warmer lower layer is large enough, a dynamical phase transition occurs. The warmer (lighter) fluid rises to the top while the colder (denser) fluid sinks, creating an ordered pattern of convection cells resembling a honeycomb when viewed from above.

Prigogine emphasized that energy flow and dissipation are the active agents in the phase transition, inducing the collective behavior that constitutes the ‘dissipative structure’. However, this phenomenon depends on the ability of the liquid to absorb and store thermal energy. It also depends on the liquid’s ability to expand as it rises. This reduces its density as it rises. In fact, it depends on the formation of a cycle.

Cycles go hand in hand with the aforementioned compartmentalization, as these two aspects are the two sides of coherence: (i) cycles, through which energy is stored in the form of long-lasting sustained excitations (released by resonance or by dissipation, see below), and (ii) compartmentalization resulting from the anisotropic distribution of matter due to confined field gradients. These are the prerequisites for the formation of cavities that enable the formation of confined states and favor new vibrational modes [3,9,64]. Thanks to coherence, there is a close relationship between function (cycles) and structure (compartments) in living matter (Figure 6).

Thanks to coherence, this fact alone is sufficient to demonstrate the deep identity between function (cycles) and structure (compartments) in living matter. This calls us to abandon the conventional, digital-like idea of “information”. The cycles and the compartmentalized structures of any living system are the real prerequisite for the high thermodynamic yield, and the capability to keep entropy close to zero. The same holds for the tight energy management. Every organism is an open system whose organization is maintained in a kind of ‘dynamical steady state’ through the flow and storage of energy and matter. If this ceases, the organism dies. However, this steady state is not a static, bulky, homogeneous phase (in a strict thermodynamic sense) enclosed within a container (as for the Bénard cells in heated water). On the contrary, within the organism, there are organized heterogeneities and mobile structures at all scales resulting from the coherences at work. There is no homogeneity, no static part fixed at any level—it is an order in time rather than in space. That is why living organisms can also be considered as nested time crystals (see Figure 7). Within a living body, there are organs, tissues, and cells. There are also vesicles, pockets, niches, ducts, intramolecular sites, folds, and so on. Each of these has a certain degree of autonomy and closure. Despite this, thanks to nestedness of coherences (acted upon a variety of degrees of freedom) they are like instruments which relate to one another so as to yield a common “music” [20,39].

For example, spatially, a single cell is divided into many compartments by membrane stacks and organelles. Each compartment has its own ‘steady states’ of processes that can respond directly to “external” stimuli and release signals to other compartments. The term ‘external’ stimuli is used here in quotes because, in the nested, coherent structure of a living organism, the environment of a small compartment is enclosed within a larger one, which is itself enclosed by even larger ones, and so on [39,67]. The smallest compartments, ‘microdomains’ can be selectively excited to give rise to local oscillations. Complexes of two or more molecules function as ‘molecular devices’ that can operate autonomously, yet coherently, without immediate reference to their surroundings. These complexes perform various tasks, such as transcribing genes, assembling proteins, ‘pumping’ ions, extracting energy from food, moving water molecules across membranes, and emitting biophotons. All these processes occur within confined nano- and micro-spaces, with time-cycles ruled by the oscillation periods of coherence operating at that scale. More importantly, the activities in all these compartments, from the microscopic to the macroscopic, are perfectly orchestrated due to the mutual dependence and nestedness of coherences. This is why the organism is endowed with a continuity and organizational closure (a self) [8]. Such an ensemble can be considered as a dynamic super-coherent liquid crystal [39]. To maintain a high thermodynamic yield, dynamic (time) closure is even more important than spatial closure because the former allows the organism to store as much energy as possible, and to use it most efficiently in cycles, i.e., with the least entropy production. In other words, the steady ‘state’ is not a single state defined by unique “encompassing” state functions, but rather a conglomerate of organized processes in space and time. The organism has an inherent space–time structure [68,69] and cannot be represented as an “instantaneous-average bulk state”. If thermodynamics were to apply to living systems, it would have to be applied at every level of compartmentation and at any instance, ultimately, even to individual molecules or coherent domains, rather than to statistical ensembles of oscillators [21,55]. This has huge practical implications, as cell culture studies that operate solely in vitro, with an oversimplified and minimized extracellular environment can only provide very limited (or even misleading) response dynamics. Mostly because the full cellular environmental repertoire is radically truncated, drastically inhibiting the exchanging of matter, energy, and environment coupling (Figure 8).

In this regard, properly a “reformulation of the second principle of thermodynamics” for living systems—whose condensed matter cannot be considered as a homogeneous bulk—had been claimed [49] (p. 35). To make this formulation compatible and applicable with single compartments or molecular devices, McClare first introduced the important concept of a characteristic time interval (time scale), *τ*, within which a system reaches equilibrium at a temperature *θ* [55]. The energies contained in the system can thus be divided into stored and thermal energies:-Thermal energies are those that exchange with each other and equilibrate throughout the system, in a time interval, ***t_e_ < τ***.-Stored energies are those that remain in a non-equilibrium distribution for a time interval ***t_s_ > τ***, either localized within the system, or such that the higher energy states are more populated than the ground states, for a given temperature ***θ***.

The most important form of stored energy is indeed coherent energy, which can be released by any type of coherence domain. Stored energy is therefore any form that does not equilibrate, or dissipate into heat within the time interval *τ*. As mentioned above in relation to coherences, the typical relaxation timescale is associated with the given level of compartmentation, which also has its own spatial scale [55]. In accordance, McClare restated the Second Principle of Thermodynamics as follows: “useful work is done by a molecular system only when one form of stored energy is converted into another”. In other words, thermalized energy is considered unavailable for work, and it is thought impossible to convert thermal energy into stored energy.

However, even thermalized energy at an “nth level” can be converted into work [49] (p. 36). Consider the characteristic space–time scale of the structure/cycle in which energy is released and/or stored. Under such premises the inadequacy of conventional thermodynamics becomes clear. It is enough to dissipate it to a “(n + 1)th level” with a relaxation time greater than the previous level (***τ_n+_*_1_* > τ_n_***). Indeed, any internal combustion engine depends precisely on thermalized energy: work is extracted from the thermal energy (Carnot cycle or other thermal loops) of combustion. McClare rightly pointed out that useful work can be done by a molecular system via the direct transfer of stored energy, but excluding thermalization. Indeed, the process of photosynthesis, on which most of life on Earth depends is an example of the direct, non-thermal absorption of photon energy. This is why typical thermodynamic calculations based on the temperature of the sun are irrelevant, [71].

However, it is necessary to consider the characteristic space–time scale of the structure/cycles in which energy is released and/or stored. Here, non-thermal energy transfer is the principal process occurring in living systems. In such a cooperative, coordinated system, energy can be directed or channelled to perform useful work, as in the case of ubiquitous molecular devices embedded in membranes across which very high electric potentials are applied (these can, in fact, act as “Maxwell’s demons” [49,55]).

The most important aspect of animated matter is that in a system with nested space–time organization (fractality), ‘thermal’ energy (rich in microstates) if produced within a small compartment, is still stored energy within a larger compartment that encompasses the former. This is because larger compartmentation levels have slower timescales than the sub-levels they encompass. This is also evident when considering coherent oscillations, which occur at higher frequencies for the microscopic and lower frequencies for the meso-/macroscopic domains. Each level of coherence has its own size (its spatial scale, given approximately by the wavelength, *λ*, of the electromagnetic mode involved), and its proper time scale (given by the resonance frequency *ω_r_* of that mode). Therefore, the characteristic time τ, suggested by McClare, can be estimated as *τ* ≈ 1/*ω_r_*. Consider for example, the different space–time scales of two types of coherence in water: on the one hand, there are electron oscillations between sp^3^ and 5d levels (wavelength of the order *λ ≈* 100 nm and (renormalized) resonant renormalized frequency *ω_r_* ≈ 5 × 10^13^ Hz, and energy gap ∆*_g_* ≈ 0.17 eV at 300 K [72]. On the other hand, there is rotation of molecular dipoles (with wavelength up to *λ ≈* 500 μm, resonance frequency *ω_r_* ≈ 5∙10^11^ Hz and energy gap ∆*_g_* ≈ 0.02 eV in the bulk, which is possibly increased when interfacial conditions are considered [73]. The microstates generated/populated within the first level, if sufficiently “mild” (so as not to disrupt coherence in the components in hierarchically higher levels) still represent ordered (stored) energy for the superordinate (larger, slower) level.

Indeed, Mae Wan Ho [67] had very perceptively suggested that a more appropriate restatement of the second principle of thermodynamics for living systems might be as follows: *useful work can be done by molecules by a direct transfer of stored energy, and thermalized energy cannot be converted to stored energy* within the same system, *the system being defined as the (spatial) extent to which thermalized energies equilibrate within a characteristic time, τ*. Ho’s refined version provides a way of defining a ‘system’ in terms of the extent of its thermal equilibration within a characteristic space–time. Therefore, a heat engine works on thermalized energy from fuel combustion because the piston performs work outside the system containing the thermalized energies of the expanding exhaust-gas. The relaxation times at the piston scale are much, much, longer than those of the expanding combustion gas molecules. This elegantly explains why their thermalisation can be used as work (i.e., as stored energy) through the compartment in which the piston acts even though the gas molecules dissipate entropic energy at their own scale.

The explicit introduction of the timescale, and hence of the space–time structure, gives rise to two quite distinct approaches to performing useful work in the most efficient way: not only slowly (i.e., at equilibrium), as defined by conventional thermodynamics, but also quickly. Both approaches are reversible and operate with maximum efficiency, as ideally no net entropy is produced within that space–time compartment [58,74]. Of course, “slow” or “quick” should be considered relatively to the timescale, *τ*, of the nested space–time domain (and cycle) in question.

A “fast” process is, for example, the release of a photon from one site and its adsorption by another, with relaxation times compatible with the inverse of the photon’s frequency. The same applies to a charge transfer to a molecule that induces conformational change (cis/trans conformation) at a rate implying a time interval between the two shape configurations of the same order of magnitude as the excited state duration of such an electric charge.

A “slow” process is one that occurs at a rate equal to or slower than the time required for all the exchanged energies to equilibrate or disperse throughout the larger system. In the case of an internal combustion engine, the mechanical energy in the form of work, generated by the displacement of the piston (macrostate) entropically thermalizes the corresponding microstates of the expanding gases. As the combustion gas expands, the entropy increases at the level of the gaseous microstates, while thermal energy at the piston level is converted into work. The characteristic space–time scales of this second (piston) level are much slower than those of the gas microstates. This is possible because combustion is compartmentalized into a chamber. Accordingly, compartmentation and closure are crucial in enabling energy exploitation from one level to another, while maintaining near-zero levels of internal entropy.

Thus, the multiplicity of nested characteristic space–time scales of the organized compartmentalized structures enables energy to be exchanged between the different levels via slow and fast pathways (as the energy that would otherwise be converted into heat, at a given small scale remains available as ordered energy at the next higher level). The amount of energy stored in the organism is huge because the level of structuring is exorbitantly high. This energy is stored in complex chemical molecules, in macromolecular conformational fluctuations, concentration gradients across membranes, in electric fields created by charge separation and viscoelastic fields due to mechanical stresses affecting the entire cell, individual proteins or tissues. Of course, energy is also stored in ubiquitous coherent states [64].

The new concept of ‘stored energy’ developed by [49] is an excellent starting point—much better than the standard ‘free energy’. In fact, ‘free energy’ cannot be defined a priori, let alone assigned to individual molecules without considering time by time their chemical–physical context. Changes in free energy cannot be defined unless we know how far the reaction is from the equilibrium. However, the classical thermodynamic notion of “equilibrium” is inapplicable in living matter, which is by definition, a system far from equilibrium. Conversely, ‘stored energy’, originally defined by McClare in terms of a characteristic time interval, has subsequently been related to a characteristic space–time domain [49,58,69]. As such, stored energy explicitly depends on the space–time domain of the processes involved. Stored energy is meaningful with respect to single molecules as well as the whole organism. An organism can be considered primarily as a coherent super-domain of energy storage, and stored energy is coherent energy capable of doing work within its specific space–time domain.

## 5. Qualities, Meaning, and Perception

Reference [8] discusses in detail why the condition of super-coherence (i.e., simplifying: the presence of several coherent levels nested within one another) allows the emergence of the self in the organism. This derives from the organizational closure and holonomy (an overall shared eigenstate of phase), which are characteristics of any living system and determine its “identity”. This property emerges out of the non-random history of quantum macroscopic states that are causally (but not only diachronically) bound to one another. The self emerges because a common (multiplexed) phase is shared by all its components, which are thus no longer “many”—because they are coherent and can no longer be defined by the number operator, $\hat{N}$. As we have seen, such an organizational closure coexists with intrinsic thermodynamic openness, implying the quanta of matter and energy in an organism to be neither fixed nor countable. Matter and energy exist as an organized, coupled flow. Therefore, fixed degrees of freedom, countable bits, q-bits or digits do not make sense [1].

Moreover, when considering the evolution of a living system over time and space, it must always be regarded in relation to its coupled environment. It is therefore nonsensical or even misleading to study it as a finite object. This coupling, provided the organism is alive, is such that (with super-coherence at work), it is never merely a reaction that can be defined by mechanistic laws and algebraic calculations. Rather, it is always a response in which the engendered state is a consequence of (i) the preceding history (from past to present: memory) and of (ii) a thermodynamic tendency (from future to present: teleology) to minimize the system’s ground level (deepening the overall energy gap) as much as possible through the non-local exploration of the phase space [9]. This distinguishes a measurement (or a mere detection), which is feasible by an inanimate device, from an experience, which is only feasible by a system in which each event implies a non-random change in the state of each representable part in a non-random way (i.e., by a living being) [8].

We term the response dynamics by which every stimulus changes the overall eigenstate of the phase of the organism, perception. It is important to distinguish this from elementary and surjective “detection”. In such a perceptive history, it becomes clear that the meaning is “what a given existential configuration implies (thermodynamically) for the system”. This emergent relational, context-dependent category is precisely neither invariant nor reducible to quantification. It is based intrinsically on qualities, i.e., the effect or the physical implication that a given state, event, or stimulus has on the organism. The change in state of a living system is what “informs” it of this implication; this is what “feeling” means physically, and it has nothing to do with linguistics. In mammals, this moment-by-moment updated configuration is known as the neurovegetative visceral map [75] and refers to the configurations of multiple body parameters (not all of which are detectable). As outlined above, the visceral map relates to the quality of such a perceptual coupling since it is the quality of the interaction, understood as its subjective meaning, that counts and acts as “sensation” (enteroception).

The perception process, based on QED super-coherence, is related to some neuroscientific topics, such as the problem of qualia and “cognition”. We can understand qualia as “analogical qualities” (i.e., how the way the perceiver feels, or how a stimulus or condition affects the perceiver) [76], and thus do not limit them to “brain-cognition” as they are also involved in elementary unicellular organisms. We could say that the category of quanta, in terms of their countability, describes the action/reaction level, while the category of qualia refers to the stimulus/response level. Conceived as particles, quanta are numerable and describe “how many” or “how much”. In this line of thought, the term “qualia” is not the plural name of (one) “qualium”. Thinking in such terms fails to capture what “quality” means and regresses back to the reductionist concept of “information”, not based on relation, configuration, or quality (thus reducing it to a label or an empty box). Quality cannot be attributed to quantity; it is precisely in qualities that the emergence of non-invariant and context-/subject-dependent meaning occurs. This is why the qualia problem—as addressed in some neuroscientific contexts—is inadequate, because it is misled by the appeal of computation [77] (see Section 3.4.1 in reference [8] for more details). We could possibly advance the idea that qualia are simply configurations within a super-coherent (living) system, ascribed to the well-defined phase of the quanta themselves (whose number must be left unknown). Therefore, they are not measurable, precisely because coherence is a condition that can be fully experienced/participated in only from within. By measurements from outside it is possible only to retrieve projections/representations of some features of it. Thus, “qualia” are essentially context- and subject-dependent qualities.

## 6. Comments on AI

From what has been said so far, we understand that what really matters is the quality of the interaction between the living system and the environment. This quality is a “variable” determined by ‘what the configuration implies for the former’, suggesting that meaning does not belong to a stimulus per se, but rather to the unique and unrepeated *relationship* experienced by the organism. This is why, when dealing with living systems, we can speak of *perception* (close to a phenomenological perspective [78])—intended as the updating of the overall state of the system at each step of evolution in/of the set (*α*, *β*) implied in the subject-environment dialectics—which is something quite different from “detection”. Detecting the biomolecules of a red wine does not mean tasting it. Measuring it does not equate to experiencing it. This is actually serious science, not fantasy [5,33]. As Lawrence Kubie pointed out, when doing science «we are constantly in danger of oversimplifying the problem so as to scale it down for mathematical treatment» (Macy Conferences between 1942 and 1953, reported in [79]). This is precisely the key problem with the so-called “Artificial Intelligence” which has nothing to do with “intelligence” (defined as knowing, understanding, experiencing, and being aware). While AI was initially conceived for military applications [80], it has gradually seeped into the wider public to finally become a daily commodity via ChatGPT et al. [81]. AI is increasingly being utilized to model and tackle chemical dynamics in biological matter. However, this is invariably executed through the same unsettling particle-based, digital-like, stereotyped representation of living dynamics that we have heretofore interrogated.

A fleet of experts has been working on protbots, like «Alphafold» [82] and «Rosettafold» [83], which have been fed with huge amounts of data (predominantly from the gigantic Uniprot database [84]) to facilitate protein folding simulations. Likewise, chemists use chembots such as «RetroBio-Cat» [85] to assist in the design of bio-catalysed chemical processes. Similarly, “RXN for Chemistry” is used to design new pathways of chemical synthesis [86], which earned the founder of AlphaFold2 (an AI model which simulates the protein steric behavior) the Nobel Prize for Chemistry in 2024 (https://www.nobelprize.org/prizes/chemistry/2024/press-release/, accessed on 23 July 2025). However, the problems are far deeper. There is a lack of a suitable epistemological stance and a consistent physical picture of how the living phase of matter emerges. This is because neither its intrinsic features (such as coherence, non-finiteness, and thermodynamic openness) nor the difference between action–reaction and stimulus–response relationships are considered. Nor is the difference between detecting and experiencing, or between quantities and qualities are taken into account. All existing AI systems are trained from the beginning on data and models that “survived” within the database and the common mindset, which is satisfied with a first quantization description of living matter, while those that did not fit the prevailing dogma have never been considered. The dominance of the ‘surviving’ models and approaches over the inaccessible, missing, or lost ones distorts the processing of data, resulting in suggestions that do not accurately reflect how phenomena may truly be understood. This phenomenon is also known in science as “selective publication” and represents a significant bias [87] with serious implications for the development of knowledge and real-life situations. Claims by proponents of AI, that their systems will be able to “read” and “understand” belong to the genre of science fiction and are based on this naïve, oversimplified view of what living beings and cognitive processes are. Reading and understanding require symbolic and representational faculties that are necessarily based on perception and experience—something that these systems simply cannot possess. What does AI know about the sensation of raindrops on the skin, about efforts, about needs, about intentions, about *experiencing* anything?

AI systems are based on mathematics and statistics and thus lack the living dynamicity of an embodied brain and everything that it entails. Human intelligence, and biological intelligence in general, cannot be reduced to the sum of the numerable physical-chemical processes generated in cells. It is much more than the sum of its parts and its “working laws” that are unceasingly self-updated by the same thing that intelligence is: relationship. Chatbots, protbots, chembots, and all other AI devices seem to work surprisingly well. This makes us believe that these models are equivalent to the human brain—an apparent self-evident fact that keeps seeping into the collective (un)conscious and mutating into a kind of shared “feeling”. As AI has not only infiltrated academia but is already interacting with the masses, it is strongly influencing the thought patterns of their users. This extremely dangerous trend leaves tech companies to decide what is (scientifically, ethically, politically) correct thinking and how to frame controversial issues. The conflicts in Ukraine and Gaza are very specific cases that illustrate this quite well: so-called “subversive elements” are identified by an AI tool known as “Lavender” [88]; this underlines the ethical dilemma, as it is no longer the individual user alone who decides, but more so the designer or manufacturer of the AI system.

Other significant issues arise from the ill-advised attempt to hybridize the living systems with the digital ones [89], or from relying on AI for jurisprudence and legal decisions [90] (including medicine) which have been programmed a priori and are not free from biases or ideologies. Rather than handing it over to the providers of this technology, such issues should be in the hands of living beings who are capable of empathy, contextualization, and feelings [91]. As “intelligent” as AI models may appear, these data-generating algorithms are ultimately nothing more than super-performing “stochastic, or merely diachronically deterministic, parrots” [91], since they lack cognitive abilities, feelings, and emotions. Above all, no algorithm can ever gain experience—something even the simplest organism possesses—so they should not be given more significance than they actually have; they are still just well-designed but dull pieces of machinery. In a parrot-like manner, AI algorithms cannot do anything but repeat and reproduce that which they have been fed, sometimes more accurately, sometimes less.

## 7. Conclusive Remarks

In this manuscript, we highlight how the common tendency to reduce problems to a form of ‘algebraic manageability’, despite being very useful—allowing us to create models of systems and predict results through simulations—creates an idealisation of reality by reducing it to a mathematised ‘toy model’. The conventional understanding of ‘information’ in biology involves reducing processes to objects and meanings (the most relevant category) to mere ‘quantity-based’ constructs (e.g., how many bits, quanta or energy and entropy are generated in a given process). This approach implies the physical inconsistency of the finiteness and closeness of living systems and the surreptitious ontological dualism between postulated hardware and software (by associating ‘bits’ with arbitrary, isolated degrees of freedom). This leads to the disregard of quality in living dynamics, i.e., how the relationships between the parts represented and descriptively distinguished are configured, and what this ‘world configuration’ implies for each ‘subject’. Meaning is not an object or a category that exists exclusively within language. In physical terms, meaning is what a given configuration of reality (i.e., a set of relationships and couplings) implies for the living system in homeostatic, thermodynamic and electrodynamic terms. Meaning is never attributable to a stimulus or situation in itself; it is not a chemical species per se that constitutes a nutrient or a toxin, but rather its contextualisation respectively to the perceiving subject.

The ‘configuration’ (associated with meanings and implications for the organism) that we propose as an alternative to the digital concept of ‘information’ is precisely the unrepeatable interplay/relationship between a living system and its reality/context at a given stage in its thermodynamic history.

Moreover, contextualisation is ‘something’ that cannot ultimately be reduced to quantities and countability. It is intrinsically analogical and, ultimately, a quality. In the same way that the shape of a triangle is related to the number ‘3’ (of its sides), where the former is a geometric quality and the latter a reduced quantity, the meaning of something (for a living system) is related to the event/stimulus. Attempting to establish an isomorphism between the aspects that can be described, spoken about or mathematized and the emerging meanings, for a biological system, re-ignites the same issues as those raised by Gregory Bateson when he alerted us to the difference between scientific/objective data and facts or events [92]. Taking measurements and partitioning a system into elements that can be represented in a language such as algebra always constitutes a simplification that obscures the qualities and meanings that are crucial for biological functioning (as any organism is a perceptive flux and an open system).

Finally, thermodynamic openness prevents us from treating living matter as hardware endowed with an arbitrary and finite number of bits (software). This makes expressions such as ‘content of information’ meaningless (or dangerous) when referring to biology and ecodynamics, as well as any similarity (or merging) between the latter and digital, artificial machines.

In this article we acknowledge that qualities are difficult to deal with because nothing can be grasped in isolation and inserted into equations or added to other ‘parts’ to provide objective data and figures. However, we demonstrate that even if computation could be performed by idealizing the system as a ‘set of bits’, significant losses would inevitably result.

We hope that this kind of sensitivity will spread among those working in science, encouraging everyone to distinguish between what is biologically alive and what is merely a stereotyped digital surrogate, despite its superior performance.

## Figures and Tables

**Figure 1 ijms-26-07319-f001:**
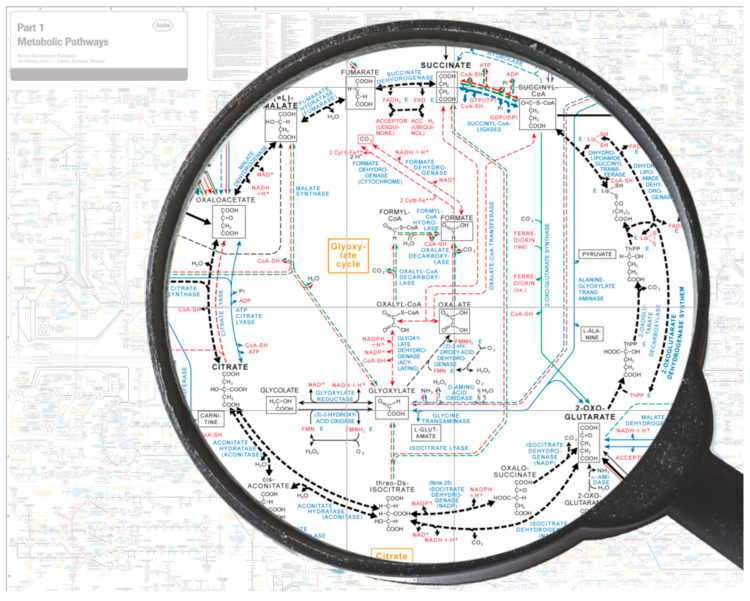
Metabolic pathways of the prokaryotic cell—highlighted: the citrate and glyoxylate cycle (modified from [27]). The reaction chain comprises a series of steps that must be completed quickly and accurately (error-free). The process, therefore, relies on intrinsic and coherently coupled electromagnetic fields that guide the co-resonant molecules (see details in Section 4).

**Figure 2 ijms-26-07319-f002:**
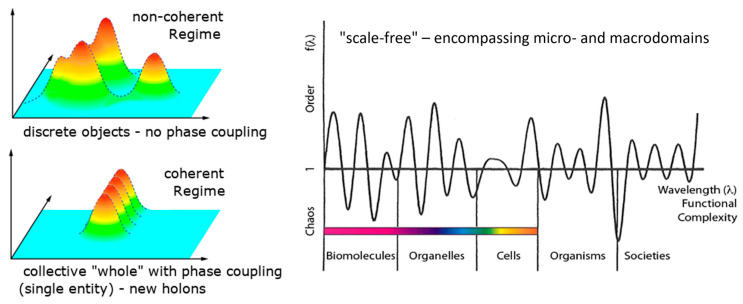
Coherence among components and oscillators (sketched as wave packets)—if their coupling and their density are sufficiently high—gives rise to a new organizational level at which response processes emerge; these are no longer reducible to the single components [29]. This emergence, which is rooted in the spontaneous breaking of certain system symmetries, can span several space–time scales. Consequently, nested super-coherence yields larger and larger coherent structures (longer wavelengths and slower frequencies), forming new *holons* [30]. The essentially scale-free nature of such dynamics is rooted in the fact that the uncertainty relation between number and phase operators does not involve Planck’s constant, enabling coherence to be established in macroscopic systems [12] (right-side image adapted from [31]).

**Figure 3 ijms-26-07319-f003:**
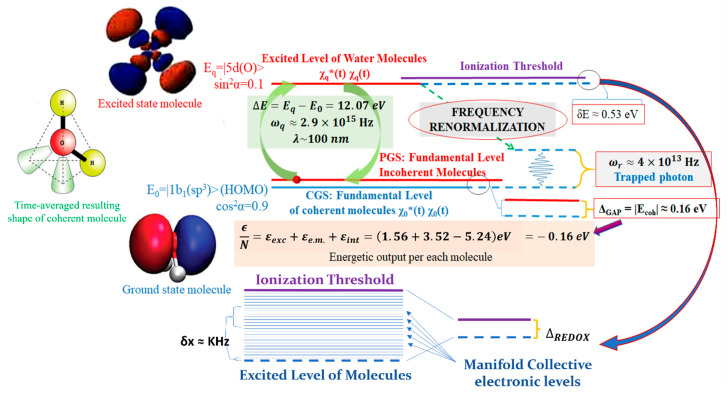
Schematic representation of the energetics of the coherent state of water molecules showing that this state results from continuous collective molecular oscillation between two states. This oscillation is driven by the self-trapped em-field, whose phase (and renormalized frequency) is locked to the phase of the coupled matter-field. In this scheme, the value of the energy gap—that is, the energy difference between the ground level of isolated molecules and the fully coherent one (Δg ≈ 0.16 eV)—refers to the latest calculation performed for liquid water, neglecting the temperature contribution. Legend: CGS, coherent ground state (bottom blue line); CES, coherent excited state (red top line prolonged into blue dashed line); PGS, perturbative ground state (bottom red line), Χ_0_(t) and Χ_q_(t) are the matter-field states (fundamental, “0”, and excited, “q”) in function of time and their own complex conjugated are denoted as Χ_0_*(t) and Χ_q_*(t), respectively. For a detailed description, see [35].

**Figure 4 ijms-26-07319-f004:**
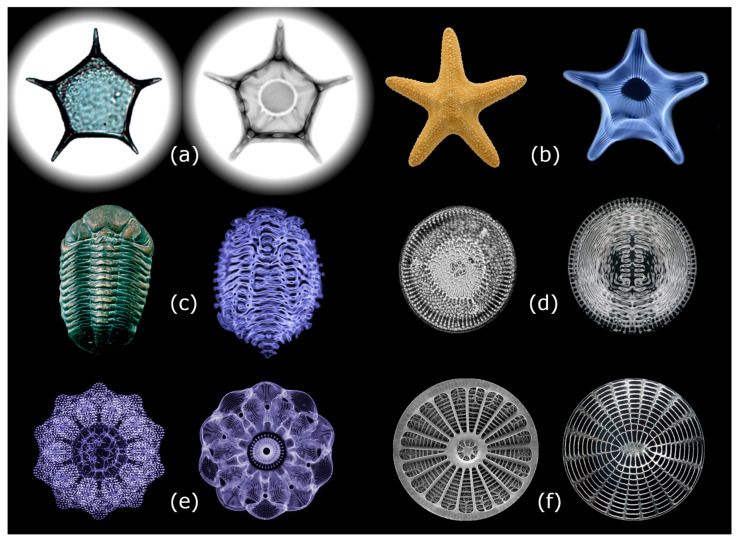
Chladni-like resonance patterns based on harmonic principles used in morphogenesis. Lower frequencies form larger structures, and increasing complexity involves adding higher harmonics. As the frequencies increase, the subdivisions become smaller and their spacings become narrower. The sequences above depict the actual biological structure, on the left, and the cymatic pattern generated with a cymascope (instrument by John S. Reid (https://cymascope.com) that renders sound visible in water), on the right: (**a**) Diatom Vollacerta Hortonil, Cretaceous; (**b**) Starfish of the Ordovician Era; (**c**) Trilobite of the early Cambrian; (**d**) Diatom Cheloniodiscus Ananinensis, Cretaceous; (**e**) DNA cross-section; (**f**) Diatom Arachnoidiscus, Jurassic (adapted from [46]).

**Figure 5 ijms-26-07319-f005:**
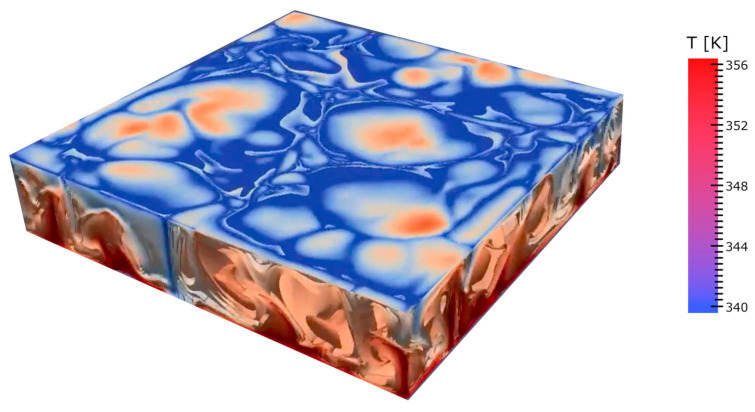
Rayleigh–Benard convection in a layer of water heated from below. Hot bottom layer, cooler top layer (available online: https://youtu.be/IBxXkpq2jWQ, accessed on 23 July 2025).

**Figure 6 ijms-26-07319-f006:**
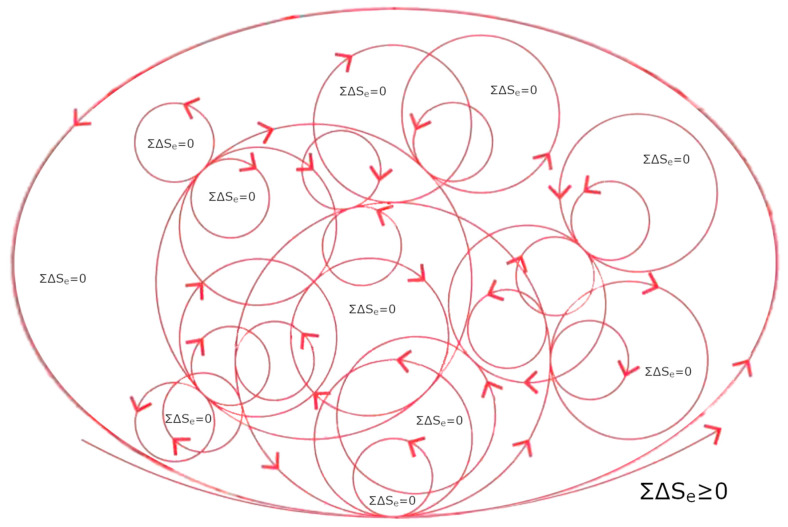
In this heuristic scheme, the life cycle of an organism (represented by the large encompassing oval) consists of a fractal structure of self-similar cycles nested within one another. Energy storage depends on the highly differentiated space–time structure of the living matter, whose coherent modes govern the different sizes spanning many orders of magnitude in space–time (all of which coupled together). The greater the number of cycles, the more energy is stored in orderly (coherent) form. Consequently, it takes longer to be thermalized, and less entropy remains within the living system (internal ΔS = 0). The average residence time of energy at a given organizational level (cycle) is a measure of the topological complexity of the system. This is a good example of how “information” in living matter is an intrinsically analogical and relational feature, possible only as configuration). The diagram schematically represents the cycles spanning all space–time scales. Based on our understanding of metabolic organization, such an energy storage system has a self-similar fractal structure and necessarily unfolds as coherent dynamics [7] (image modified from [65]).

**Figure 7 ijms-26-07319-f007:**
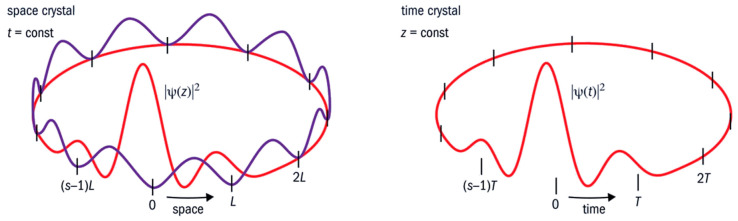
A time crystal can be considered an interacting many-body system that exhibits long-term oscillations without the need for a constant source of energy. It is best described as a stable, conservative—yet quasi-periodic and therefore evolving—macroscopic clock. The sketch on the left represents a 1D space crystal with periodic boundary conditions (violet line) and a time crystal in Anderson localization (red line). To switch from a space crystal to a time crystal, the role of space, *z*, and time, *t*, must be exchanged. This means fixing the position in space while observing if the probability of detecting an excitation at this point in space, |Ψ(*t*)|^2^, is Anderson-localized around a certain moment of time (image reported from [66]).

**Figure 8 ijms-26-07319-f008:**
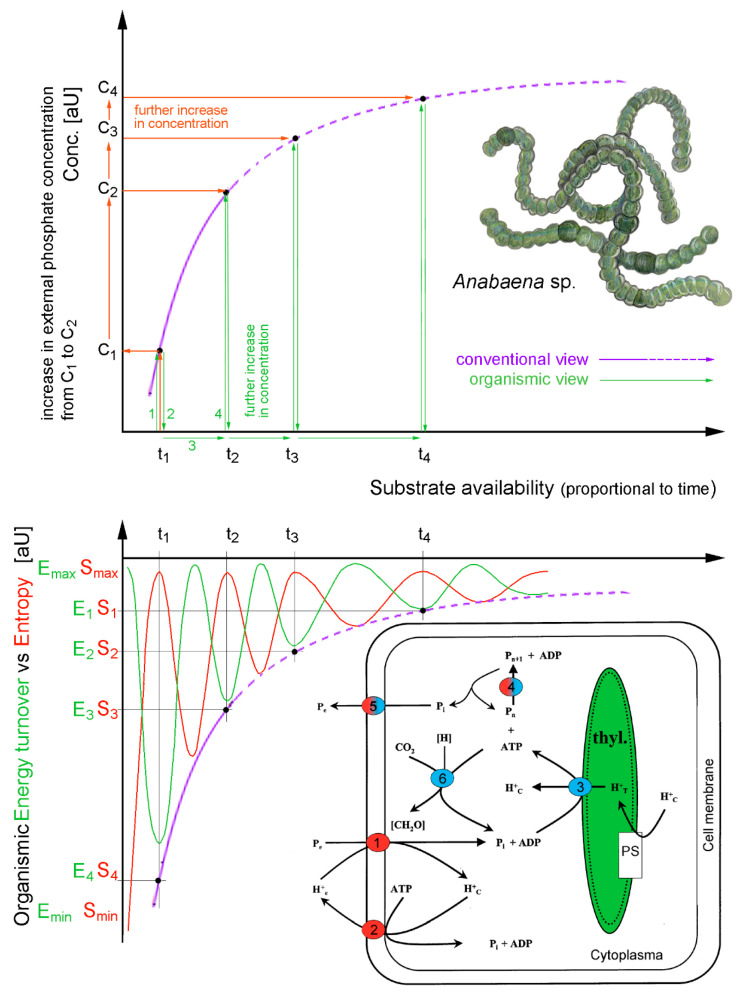
Adaptational metabolic sequences in *Anabaena* sp. (**Top panel**): along the time axis, the phosphate concentration of the external environment (around the *Anabaena*) increases at times t_1_, t_2_, t_3_, t_4_ (shifting from C_1_ to C_2_ at t_1_, from C_2_ to C_3_ at t_2_, and from C_3_ to C_4_ at t_3_). Unlike what could be observed from outside without considering the organizational complexity of living matter, the organism does not simply engage in a gradual adaptation of its metabolism (violet trendline). On the contrary, to cope with each variation in phosphate concentration, *Anabaena* resets its metabolism completely, which is suspended before restarting on an updated regime (as depicted by the green arrows in the top panel, under the violet line). Of course, the greater the increase in phosphate concentration (threatening the biochemical equilibrium of *Anabaena*), the smaller the remaining coping range is. (**Bottom panel**): the new configuration that such a phosphate stimulus implies, forces the organism to boost the uptake reaction kinetics during the time slots of t_1–2_, t_2–3_, t_3–4_. During these transitions, *Anabaena*’s energetic effort rises, pushing the organism toward a temporary entropy maximum [70]. This discontinuity in maintaining internal homeostasis at each given phosphate concentration is backed by the trend of the organism’s internal entropy (red line), which shows a temporary increase at each chemical “offence”. This is minimized once the organism has completed its organizational update. This update can be understood as a rearrangement of the work-frequencies of (some) super-coherence domains, which are ascribed to the biochemical cycles that need to be reset in order to maintain coherence and functioning. As is evident, with each increase in phosphate concentration, the restored minimal entropy is higher than before, showing that the organism’s homeostasis (coherence, energy gap) is becoming weaker. Thus, we observe a violet “saturation trend” up to the level at which entropy cannot be minimized any further, and the organism dies due to phosphate overloading (poisoning). Another interesting aspect regards the fact that the variation in internal entropy is opposite to the metabolic energy turnover level (green line), which is chemically detailed in the inset box. This provides further evidence that, in living matter, “information” cannot consistently be interpreted as is usually done in the context of IT (*à la* Shannon), as it is the opposite of entropy. Moreover, it cannot be considered something that is added to the living matter, as it is indeed its configuration, functioning, and structure that result from the coupling with the environment (context). The living system is not a physical object, but an open flow and at the same time a response–process. See the text for an in-depth analysis.

## Data Availability

Not applicable.

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
