# Peer review of "From “Information” to Configuration and Meaning: In Living Systems, the Structure Is the Function"

_ijms, 2025, doi:10.3390/ijms26157319_

Round 1

Reviewer 1 Report

Comments and Suggestions for Authors

This article challenges the traditional concept of “information” in biological systems. The author argues that life systems should be understood from the perspectives of “configuration,” “meaning” (defined as the thermodynamic impact of real-world configurations on organisms), and “properties.” These perspectives are grounded in the principles of quantum field theory (QFT) and quantum electrodynamics (QED), with particular emphasis on concepts such as coherence, symmetry breaking, and dissipation. The core argument is that “structure is function,” and this relationship is deeply interconnected, dynamically evolving, and context-dependent. By synthesizing complex ideas from physics, philosophy of science, and biology, the article critiques the application of traditional reductionism and information theory in biology and proposes a more holistic, process-oriented framework, thereby making a significant contribution. However, the article requires further improvement in terms of readability, empirical support, and content coherence. The following are some suggestions I offer, hoping the author will consider these improvements before resubmitting the paper:

  1. Although the author does provide explanations, many concepts related to QFT and QED remain obscure and difficult to understand for readers without a physics background. The authors should consider: adding a brief, easy-to-understand introductory section or supplementary material explaining the key concepts used and why they are particularly applicable to biological systems.
  2. The paper relies heavily on theoretical derivations and lacks experimental validation. It is recommended to include specific examples of QFT applications in biological systems (e.g., experimental evidence or computational simulation results related to biological water coherence) to enhance persuasiveness.
  3. The article contains some spelling/grammar errors. A final thorough proofreading is recommended. For example: Line 30: “emerges im-plying” -> “emerges, implying”; Line 344: “interplay ow a water connectome” -> “interplay of a water connectome.”
  4. Section 6, “Comments about AI,” is weakly connected to the core theme, and critical discussions (such as AI's “selective bias”) do not form a logical closure with the QFT framework, appearing out of place. It is recommended to delete or focus on the limitations of current AI methods in biology.

Author Response

Authors’ Response to Point 1:

Thank you for your valuable suggestion, we are grateful that you have appreciated the content and aim of our manuscript. Regarding some more “in-depth” concepts of QFT and QED we were relied on directing the reader to the quoted literature, including our two previous papers (as reference [29] and reference [33]). 

Nonetheless we think a brief explanation at this point is helpful.  In 1957, when biologist Albert Szent-Györgyi (Nobel laureate) wrote his book Bioenergetics,[1] he argued that biologists had considered almost all of the important components of living matter, except for two: Water and electromagnetic fields.  He emphasised that the whole trend and the whole picture of chemical reactions in living matter remained unexplained unless a central chemical problem was solved.  The majority of biochemical reactions are oxidoreductive.  What does that mean?  In physical terms, this implies that one molecule releases an electron and another molecule takes it up – in essence a chemical reaction based on electron transfer.  For this to happen, an electron must be available.  So, what are the molecules that willingly give up electrons?  While oxygen is a typical oxidiser, it readily takes electrons, while metals are typical species that readily donate electrons.  Metals, however, are present in living matter only in very small, almost imperceptible quantities.  Therefore, they cannot account for the flow of electrons that occurs every time an oxidation-reduction reaction takes place.  The presence of free radicals in the body simply indicates that the system sometimes does not complete oxidation-reduction reactions and fermentative processes develop – as outlined by Otto Warburg already in the 1920s.[2] 

So, apart from the scarce metal ions available, who else could act as the electron donor?  According to Szent-Györgyi: Water!  However, in the isolated water molecule, it takes 12.6 eV to remove an electron (12.6 eV is the equivalent of the energy released in a collision between molecules at a temperature of 145 k°C).  This amount of energy can’t be provided by enzymatic activity.  Proteins are rather dielectrics not conductors. As Szent-Györgyi noted, there is something mysterious about it.  The Krebs cycle, for instance, is a typical example where protons and electrons are paramount to make it work.  Biochemistry does not provide the answer and Szent-Györgyi was aware of that.  In fact, he noted that, based on experimental evidence, there is practically no “free” space in living matter.  The cell is a crowded space.  This means that the water that makes up our bodies — 70 % by weight but 98 % by molar concentration.  Accordingly, to him intracellular water must be a special kind of water.  In the context of the living state, water molecules behave differently when they form an intermolecular organisation than when they are isolated.  Thus, an isolated water molecule requires 12.6 eV to give up an electron.  However, this changes within the living matrix when the water molecule is surrounded by many other water molecules or molecules of a different nature.

An atom is approximately one angstrom in size.  When an atom changes configuration, an energy jump occurs that requires a certain amount of energy.  According to quantum physics, a photon able to excite an atom is essentially a wave.  Given that it requires 12.6 eV to set free an electron from water (to be precise from the oxygen atom of the water molecule),[3] the associated wavelength of the photon corresponds to approx. 1000 angstroms (~100 nm).  This means that the photon is an object a thousand times larger than the water molecule.  But this photon, with a wavelength of 100 nm, is therefore able to interact with more than just one molecule at a time (a cube measuring 100 nm at physiological temperatures, contains about >20,000 molecules.  As it is known that the vacuum continually emits (virtual) photons, this train of photons captured by water molecules (and immediately released till their volume numerical density is below a given threshold). Once the molecules in that given volume are many enough (as to reach about 0.32 gr/cm3) the photons cannot be released back to vacuum anymore and stay self-trapped in the matter ensemble. This transduces into an oscillation between a state in which the electrons are tightly bound (sp3 orbital) and a state in which the electrons are almost free (12.07 eV, the 5d orbital).[4]  This implies that only a fraction of an eV (about 12.6 – 12.1 ~ 0.5 eV) is needed to ionize the oxygen atom within a water molecule.  Moreover, because the molecules in this volume (about the wavelength cube) exceed a critical threshold density (liquid state) they form a collective system (termed coherence domain, CD – which, along with the vapour-like interstitial water, forms a biphasic system).  With a biochemical reaction providing something like 0.5 eV to the system, it is easy to imagine that water in this why turns from an insulator into a conductor.  With almost free electrons (about 6·105 per CD) present, the whole chain of redox reactions in chemical cycles is finally possible.

In section 3, we provided an overview of essential concepts needed to explain why we cannot speak of “information” in the conventional sense. Therein, we summarized the arguments which explain the limits of the conventional idea of “information”, unapplicable in biology and here in the responses we provided the synopsis of how coherence is the only possible condition for biological matter to function.

Authors’ Response to Point 2

Providing experimental validation of this methodological reframe is not feasible because the focus here is on offering an alternative approach to studying life dynamics, while avoiding inconsistent (and dangerous) reductionisms. However, ref. [29][5] and [35][6], as well as the work by other researchers, provide results and/or theoretical achievements that validate biological coherence and the role of a compartmentalized connectome of (super)coherent water in living matter.  

There are numerous practical examples in biology – a summary paper gathering these will appear shortly.[7]  Briefly some highlights of this upcoming paper: oscillatory modes of microtubules and the associated potential gradient inducing coherent alignment of the associated aqueous phase; and another crucial property regarding protein de/activation induced via coherent/decoherent water alignment (on given degrees of freedom).  Furthermore, transmembrane proteins are incorrectly viewed as molecular shuffling devices whereas they actually operate on principles involving quantum tunnelling.  Similarly, neurotransmitter activity is not a diffusion-like mechanism but rather a transmission process involving wave packets. Photosynthesis has long been recognised as a coherently coupled photon-capturing device, and the striking similarities to solid-state physics are staggering.  At the cellular level, each cell is considered a cavity resonator in the context of quantum biology, with distinctive resonance modes that give each cell group its identity. This property is changed/tuned during mitosis, and the associated epigenetic modulation is reacquired electromagnetically at the final stage of the mitotic sequence to prevent uncontrolled cell proliferation. 

Already Fröhlich stipulated that biological systems will exhibit long-range phase correlations enabling that individual dipolar units to merge into an oscillating ensemble acting as a single giant dipole.[8]  At the cellular level this regards (i) the pairing of homologous chromosomes in meiosis,[9] (ii) enzyme-substrate attraction, helping other enzyme molecules to become metastable, which in turn attract additional substrates.[10]  The energy liberated by the chemical processes acts as an em-coupling agent, resulting in accelerated enzyme activity until most of the enzyme (being metastable) are locked, after which the process continues in a normal way;[11] and (iii) the control of cell division, which is decisive in the cancer process;[12] i.e. two electrically neutral cells can attract each other over a distance much larger than the range of the usually considered van der Waals forces, provided they are properly tuned to a common, in-phase oscillation that translates into a net force operating over distances (as we described in section 4 of the present paper) of the order of 10 mm.[13]

The coherent dynamics in biological systems[14] is behind the orchestrating biochemical reaction dynamics.[15]  Such electromagnetically directed biochemistry brings about another phenomenon known as the "fundamental uncertainty relationship"[16] and its pivotal role in maintaining a well-balanced biochemistry (operating at low energetic requirements, with little waste) that is fundamental for the vital state of organisms.[17]  Said that it becomes clear how geomagnetic influences coupled to very weak alternating extremely low frequency fields (ion cyclotron resonance) cannot but affect biota.[18]

All of these kind of processes, in the name of coherence, cannot be reduced to a discrete-digital like concept of “information”, right because they pertain (i) to non-defined number of degrees of freedom (as coherence requires), and (ii) to relational dynamics, context dependent and qualitative nature (in the sense that concerns what the configuration of reality implies for the homeostasis of an organism/species/ecosystem in that unrepeatable ‘coordinates’ depending on history and on teleology; exactly like anybody can measure the wavelength of a photon, but no one can report the quality, the effect of seeing “red”, which is a quality… in fact nobody knows how a bee perceives the light at a wavelength equal to 632 nm, which we see as “red”).

We cannot perform a computational simulation because we are dealing with qualities and analogical aspects such as meanings, context-depending configurations and history-depending series of work-frequency sets (in the hierarchy of nested coherences). Moreover, we should not perform a computational simulation to mimic the living dynamics right because by doing that we would need to arbitrarily define which degrees of freedom count as “bits” and we could never deal with qualities. In the end we would stereotypy the living system as an oversimplistic collection of 0/1 bits.  Despite not being straightforwardly computable, these qualitative and analogical aspects are decisive for living syntax.

Authors’ Response to Point 3

We corrected them and checked the phrasing.

Authors’ Response to Point 4

The comment about AI, and its increasing use in life sciences and biology in general is actually related to a significant problem: all the databases and algorithms used to develop AI platforms adhere to the “first quantization” paradigm that regards conception of molecules, water (reduced to H2O), living matter based on QM and the idea that information can be reduced to bits and quantifiable degrees of freedom.  This will exacerbate the existing bias in future scientific literature. However, in that section, we try to restrict the range of our comments to this topic mainly.

[1] Szent-Györgyi A (1957) Bioenergetics. Academic Press, New York

[2] Warburg O. The Oxygen-Transferring Ferment of Respiration. Nobel Prize Lecture (1931): https://www.nobelprize.org/uploads/2018/06/warburg-lecture.pdf

[3] Renati P, Madl P (2024) What Is the "Hydrogen Bond"? A QFT-QED Perspective. Int J Mol Sci. 25(7):3846. doi:  10.3390/ijms25073846

[4] Preparata G (1995) QED Coherence in Matter, Singapore: World Scientific. doi: 10.1142/2738

[5] Madl P, Renati P (2023) Quantum Electrodynamics Coherence and Hormesis: Foundations of Quantum Biology. IJMS 24(18):14003. doi:  10.3390/ijms241814003

[6] Renati P, Madl P (2024) What Is the “Hydrogen Bond”? A QFT-QED Perspective, IJMS 25(7): 3846. doi:  10.3390/ijms25073846

[7] Madl P (2025) Quantum Biology.  In: Martel J (ed) Recent Advances in Bioelectromagnetism. Academic Press (in press). https://play.google.com/store/books/details/Recent_Advances_in_Bioelectromagnetism_Implication?id=dKVEEQAAQBAJ&hl=en-US

[8] Fröhlich H (1972) Selective Long-Range Dispersion Forces Between Large Systems, Phys Letters, Ser A 39, pp. 153–154. doi: 10.1016/0375-9601(72)91060-2

[9] Holland, B.W. (1972) Dynamic specificity and the pairing of homologous chromosomes in meiosis. J Theor BioI, 35, 395-397. doi: 10.1016/0022-5193(72)90047-1

[10] Fröhlich H (1975) The extraordinary dielectric properties of biological materials and the action of enzymes. PNAS, 72(11), pp. 4211-4215. doi: 10.1073/pnas.72.11.4211

[11] Fröhlich H (1970) Long range coherence and the action of enzymes. Nature, 228(5276), 1093. doi: 10.1038/2281093a0

[12] Fröhlich H (1978) Coherent electrical Vibration in biological systems, and the Cancer problem. IEEE Trans Microwave Theory Techn, 26, 613-617.doi: 10.1109/TMTT.1978.1129446

[13] Hyland GJ (2009) Fröhlich's Coherent Excitations & The Cancer Problem—A Retrospective Overview of His Guiding Philosophy, Electromag Biol Med, 28:3, 316-329. doi: 10.3109/15368370802708827

[14] Buzzacchi M, Del Giudice E, Preparata G (2002) Coherence of The Glassy State. Int J Mod Phys B, 16(25), 3771–3786. doi: 10.1142/s0217979202012116 

[15] Del Giudice E (2004) The psycho-emotional-physical unity of living organisms as an outcome of quantum physics. In: Globus GG, Pribram K, Vitiello G (eds) Brain and Being. John Benjamin Publ. Amsterdam (NL). doi: 10.1075/aicr.58.06giu

[16] Madl P, Renati P (2023) Quantum Electrodynamics Coherence and Hormesis: Foundations of Quantum Biology. Int J Mol Sci. 24(18):14003. doi:  10.3390/ijms241814003

[17] Dey, D., Tiwari, A.K. (2020) Controlling Chemical Reactions with Laser Pulses. ACS Omega, 5(29): 17857-17867. doi: 10.1021/acsomega.0c02098

[18] Del Giudice E, Giuliani L (2010) Coherence in Water and the kT Problem in Living Matter. In Giuliani L, Soffritti M (eds) Non-Thermal Effects and Mechanisms of Interaction between Electromagnetic Fields and Matter, Eu J Onc, 5, 7-23. Fidenza: ICEMS-Ramazzini. ISBN 978-88-6261-166-4. http://www.teslabel.be/PDF/ICEMS_Monograph_2010.pdf

Reviewer 2 Report

Comments and Suggestions for Authors

Honestly, I'm quite confused about the form and the materials of this manuscript. While the manuscript is submitted as a 'Review', the form of the manuscript is not quite like a 'review'. First of all, I do not actually know the core topic that the paper would like to review. Furthermore, I do not see typical high volumes of citations appearing in a review paper. To me, this manuscript is not like a review paper, but even worse, it is not like a normal research article either. There is no solid foundation, introduction, main results, discussion, etc. Overall, it looks like a science popularizing article and contains no solid results. Of course, I understand the living systems are so complicated that people can only use a hierarchy of accuracies to describe them, but at the end of the day, I don't know what the authors would like to convey.

Author Response

Authors’ Response

Dear reviewer, we understand your perplexity and concerns perfectly. We are aware about the fact that this manuscript is rather unconventional and may seem somewhat “unsound”.  The core issue is that certain fundamental assumptions, developed over decades, have been found to be highly biased, and some have even been found to be incorrect.  This has led to the misconception that contemporary science no longer questions these incompatible understandings that shape, metaphorically speaking, the 'trunk of the tree of knowledge'.  Unfortunately, contemporary science is preoccupied with filling in the gaps in the 'canopy of the tree of knowledge'.  However, this endeavour is destined to fail unless the fundamentals are properly adjusted – that is “trimming the tree of knowledge”.  To illustrate this point, we can provide the following examples:

  • The Arndt-Schulz principles in chemistry, tied with the Weber-Fechner principle, paved the way for electromagnetic hormesis.[1] Thus, radiation hormesis in the ionizing range of the electromagnetic spectrum was extended to include most of the non-ionizing part as well. As will be outlined further in the following paragraph, EM interactions are the primary means by which suitable biomolecules can interact over long distances.  Indeed, QED clearly demonstrates how action at a distance is possible not via particle-particle interaction, but rather via the oscillatory properties of their fields.[2]
  • The average adult is 70% water by mass, but 98% by number (molar concentration). Today, biochemistry knows almost everything about the 2 % of biomolecules, but hardly anything about the 98% majority and why water is so predominant. Recent advances in coherent QED have revealed that liquid water is biphasic, consisting of a coherent and a non-coherent fraction.[3] While biochemistry only considers the bulk as non-coherent, pure solvent, the coherent fraction is special and constituting the whole matrix/connectome in biomatter, as there are many interfaces in the form of membranes, microtubules, organelles and proteins etc., that align water molecules and leave hardly any space for incoherent phase water.[4]  On the periphery of such coherent regions of water, there is a set of almost free electrons that can oscillate in various levels, engaging in an orchestrated and directed manner in biochemical reactions that are highly targeted and almost error-free (as we explained in section 4 of the present paper).  This enables nested coherences to be established, which manifest as biological complexity, i.e. organization into cells, tissues, organs, organisms, ecosystems and biomes, and even, possibly (if we only followed biological syntax), society as a whole.[5]
  • from there it follows that nature, rather than being competitive, is a highly orchestrated cooperative network.[6] The dominance of the mechanistic interpretation originating from Galileo-Newtonian perspective - is built on its own principles, which are not those of biology but those of Machiavellian economics and the classical (Descartes’s) dualism between structure and functions.  The latters require competition, while the former is cooperation based on coupling of resonance modes and phase correlations.  In this sense, economics (the way it’s been built up) is inherently a pathological fact that breeds disease.  Thus, human evolution is still rooted in prehistory, because the human species, has not had the chance to unfold properly.  In order to properly develop, the members of the human species must resonate with one another.  Has this happened in human history yet?  No, as humanity is still indoctrinated with the principles related to economics, competition, division in parts and reduction to quantities/values (in science too).  In fact, competition is the opposite of resonance.  How can one resonate with someone if he or she has to be careful not to be outcompeted?  How can one resonate with someone else if individuals have to prove their supremacy?  So, in this sense, as long as there is a regime based on competition between human beings, the problem of health and wellbeing can never be solved.  Medical staff, therapists, psychologists may do all they can, but their results will always be temporary.  Their poor patient – so to speak is dismissed from hospital or psychotherapy just to discover that he or she became redundant and is fired.  At this point, all the work done to curing work has been in vain.

These few examples have such huge implications that it would be unsuitable to build on the classical structure of our paper, as these novel pieces of knowledge call into question many of the presumptions that were previously considered to be the sole truth. We therefore need to first establish the fundamentals and build on them. 

It is for this reason that we could not to follow the traditional structure of a manuscript, in which data are presented and discussed, as the topic concerns theoretical reflections on a problem in the method of science in dealing with living systems.

What we tried to convey is the need to include aspects that are not measurable or computable (but are physically grounded), such as qualities and meanings, in scientific methodology.  This is because as the second quantization (QFT) approach demonstrates, living systems are essentially “processes” rather than “finite and isolated objects”.  They cannot be addressed as systems with an “information content” because they are subject to intrinsic openness and becomingness.  The information is not something that can be worn (or injected) like a dress (or a signal); rather, it is the qualitative condition of reality that implies effects of coherence (thermodynamic stability) of the organism moment by moment, with the states that succeed one another being reciprocally dependent (both from past to future, memory, and vice versa, teleology).

To achieve this, we must shift from a QM-view to a field formalism and abandon the “counting” of the number of oscillators (obeying the phase-number complementarity/uncertainty relationship: ΔN·Δφ ≥ ½). This implies deconstructing any “digital-like” or computational conception of “information” (based on quantity) and therefore compels us towards a wider logical openness.  In accordance with the phenomenologist M. Merleau-Ponty,[7] this led us towards a new model of rationality able to deal with unrepeatability, quality, meaning, and openness.  The manuscript is thus devoted to reviewing the arguments that support this perspective.  We think that, more than ever before, this paradigm shift is urgent in order to counteract the increasingly dystopian and bio-technological attitude with which contemporary science identified.  However, you are probably right that this manuscript should not be categorized as a “review”.  If you or the editors could provide valuable feedback about the most suitable typology to assign to the work, we would greatly appreciate it.

[1] Madl P (2024) Ultra-Weak Electromagnetic Hormesis as the Baseline of Gurwich’s Work. Saratov Fall Meeting, (RUS), doi:  10.24412/cl-37275-2024-1-150-183

[2] Henry M (2016) L’Eau et La Physique Quantique.  Dangles ed. Escalquens (FRA). ISBN 978-2-7033-1147-8

[3] Renati P, Kovacs Z, De Ninno A, Tsenkova R (2019) Temperature Dependence Analysis of the NIR spectra of Liquid Water Confirm the Existence of Two Phases, One of Which is in a Coherent State. J Molec Liq 292: 111449, doi: 10.1016/j.molliq.2019.111449 0167-7322

[4] Renati P (2024) QED Coherence in Condensed and Living Matter. Verlag Kovac, Hamburg (FRG).  ISBN 978-3-339-13994-8

[5] Madl P (2025) Quantum Biology.  In: Martel J (ed) Recent Advances in Bioelectromagnetism. Academic Press (in press). https://play.google.com/store/books/details/Recent_Advances_in_Bioelectromagnetism_Implication?id=dKVEEQAAQBAJ&hl=en-US

[6] Capra F (1996) The Web of Life – A New Scientific Understanding of Living Systems. Anchor Books, New York. ISBN 0-385-47675-2

[7] Barrett C (1987) Merleau-Ponty and the Phenomenology of Perception.  Royal Institute of Philosophy Lecture Series, 21, pp 123-139.  doi:  10.1017/S0957042X00003527

Reviewer 3 Report

Comments and Suggestions for Authors

While the manuscript critically examines the application of classical Shannon information theory to biological systems, highlighting its limitations in capturing the relational, contextual, and meaningful aspects of living organisms, there are notable problems and limitations across several key dimensions.

  1. The manuscript revisits known critiques of Shannon information theory when applied to biology, emphasizing analogical, relational, and quantum field perspectives. While these ideas have been discussed in interdisciplinary and theoretical biology literature for decades, the paper does not clearly articulate what new insights, models, or testable hypotheses it brings to the table. It largely recycles familiar critiques without novel theoretical or empirical advances.
  2. The manuscript often mixes scientific terminology (entropy, thermodynamics, coherence) with philosophical concepts (meaning, logos, ontological indivisibility) without clearly defining terms or distinguishing speculative ideas from established science. This makes the argument hard to follow and undermines logical coherence.
  3. The manuscript criticizes the notion of information as “bits” but then introduces terms like “meaning” or “configuration” without rigorous definitions or operational clarity. For example, saying “meaning is what the reality configuration implies” is vague and untestable. The argument risks circularity: information is not bits but relational meaning, and meaning is defined relationally without independent criteria.
  4. The critique of reductionism and mechanistic biology is justified philosophically but the paper assumes a holistic quantum field-based model without explaining why or how it solves the highlighted problems better than other frameworks (e.g., systems biology, network biology). The leap from quantum coherence to biological function is asserted rather than demonstrated.
  5. Many sentences are overly long, packed with multiple ideas, which reduces readability and accessibility. In addition, Minor issues in punctuation, awkward phrasing, and inconsistent tense usage appear throughout, detracting from professionalism.

Author Response

Authors’ Response to Point 1:

It’s good first to give a short recap of recent history of the “information” concept. According to John Wheeler, information is the basis of all reality.[1]  Like Wheeler during the first period of his scientific life, most biologists today still believe that all vital phenomena are material in nature. If we stay at this primary level of classical corpuscular physics, we find the lock-and-key paradigm enunciated by chemist Emil Fischer who the foundation of all molecular biology,[2]  Anyone with a physics background, knows that there are invisible fields underlying immaterial waves – foremost the electromagnetic field. It's therefore astonishing that so many biologists still find it hard to believe that an immaterial electromagnetic wave can act on a receptor in the absence of any substrate.  Shannon paved the way for what is now known as information theory.[3] Instead of looking at the cost of measurement, Shannon asked the question of the cost of communication.  For Shannon, information really has no meaning.  By just focusing on a reliable data transmission rate, it is quite clear that Shannon isn't really concerned with information.  It's a theory of the transmission of information, not a theory of its meaning.  Serious problems even arise from the kinetic information theory invented by Norbert Wiener,[4] in which the quantity of information in a system is a measure of its degree of organization, with entropy being a measure of disorganization. This led Brillouin to introduce the notion of neg-entropy,[5] i.e. changing the sign of Shannon's entropy to be consistent with Wiener's definition.  What followed are decades of misunderstanding and confusion ensued from this simple change of sign.  Shannon's information is in fact entropy and corresponds to a number of choices, a number of microstates, an indeterminacy. Brillouin changes the sign, and information suddenly becomes synonymous with order, but this is not the way we intend to look at the concept of information.

To our knowledge, till now, these ideas have been inadequately discussed in interdisciplinary and theoretical biology literature … in the sense that:

this issue mainly arises in exquisitely philosophical contexts, rarely in theoretical biology, and never in the perspective of coherence or in a QFT descriptive approach.  The latter demonstrates the dynamical emergence of order through symmetry breaking and overcoming von Newmann theorem.  In doing so, it is possible to overcome the dualistic/cartesian perspective, whereby structure and the function are considered separate entities. 

This allowed limiting (and even complementary) views, according to which living matter

  • on the one hand is comprehensible about its behaviour by digging down, zooming in, into its microscopical constituents, and
  • on the other hand – this procedure being de facto unable to justify the features abundantly listed in the manuscript – needs the addition of “something else” like “information”.

In the context of “hard sciences” the latter term is eventually pretended and postulated (more or less explicitly) to be reduced through a discrete, quantifiable array of arbitrary states/bits/units.  This ignores the question of how these “bits” are decided and defined.  This raises an even more profound problem: the idea that “information” is a category/observable based on quantity and subject to computation.

To be sincere, based on what we can observe in contemporary scientific literature (from physics, to theoretical biology and even the philosophy of science), it seems that these problems are not at all understood and that these ideas are actually completely ignored.

In this manuscript we aim to address and highlight this fundamental problem, which has so far not been considered a problem by most scientists, on a sound physical framework (despite being methodological and, of course, NOT computational). We find the ideas introduced here, especially in terms of their perspective and relationships to other themes, to be very novel and meaningful.

Authors’ Response to Point 2 and 3:

We clearly understand this critique and agree with it to some extent.  However, it deals precisely with the core of the problem we are addressing: until the hard sciences believe that “defining something” in a mathematizable way restores the true status of reality, and that this kind of rationality is the only possible one, an incomplete and misleading knowledge of the living state will continue to be perpetuated and validated.  Opening up the methodology and vocabulary to include analogical aspects and qualities without being “illogical” or “irrational” is the first step in tackling the non-finiteness and relational essence of living dynamics and their dependence on “semantic”, qualitative, coordinates.  We are fully aware of the “philosophical” nature of these claims, but we must start somewhere…

We could understand that saying “meaning is what the reality configuration implies” is vague and untestable.  The point is that, in living matter, “what the configuration of reality implies for an organism” (only in that very moment, place, context) is exactly… “untestable”, and we’re therefore suggesting to raise such an awareness.  Let us use this metaphor: a dog’s third time smelling a person is not equal to the first nor to the eleventh. We think, and we need to come to terms with this uncomfortable fact, and with the need to adopt non-computational language to express that.

In this regard, it’s noteworthy to mention the historical work of the chemist Giorgio Piccardi and his colleagues, who recorded changes in the chemical-physical behaviours of their test systems (e.g. the shape and the size of precipitated crystals from a solution).  Depending on the local conditions (environmental configuration) the observed results allowed him to invent and build the so-called “storm-glass” (a sealed glass tube, containing a hydro-alcoholic solution of ammonium chloride, potassium nitrate and camphor), which acts as a “weather forecasting tool”.  Rather than working on humidity or temperature variations, it relies on the environmental energy configurations.[6]  His work led him to conclude that “heterogeneous, out-of-equilibrium open systems, if sufficiently complex, exhibit dependencies on any external stimuli even infinitesimal in energy”.[7] Among these stimuli, he was able to demonstrate that among the affecting stimuli there were: i) the anisotropies of the geomagnetic field, which vary from place to place over time; ii) super-long wavelength and extremely low amplitude electromagnetic waves, such as Schuman modes; iii) meteorological, astronomical and cosmological events, such as weather, the changing of the seasons, sunspots cycles, the phases of the moon and planetary alignments.  These events do not provide any measurable variation, except for a few qualitative aspects sometimes, and change over time (often being not repeatable within human-scale timelapses). This last point is decisive because it fundamentally questions and criticizes the notion that “repeatability” and “reproducibility” are prerequisites for a scientific study.  Conversely, in thermodynamics in particular, neglecting the fact that the history plays a key role in how a system behaves is right what prevents us from understanding those meaningful, unmeasurable, variables that matter in the dynamics of life.  Piccardi was one of the first (hard) scientists to advocate a methodological revision in the life sciences.

We therefore state “the study of fluctuating phenomena requires a new research methodology and different criteria for scientific verifiability as research often refers to the study of contexts that are not manipulable by humans.  This makes it impossible to use the reproducibility of an observed phenomenon as the sole and unique criterion for scientific truthfulness.  

Quoting G. Piccardi:[8]

«In the case of fluctuating phenomena, as the conditions under which a phenomenon occurs cannot be fixed, we cannot consider as valid only those experiments that produce consistent results when performed under identical traditional conditions.  Therefore, when it comes to fluctuating phenomena, we must accept as valid (also) those experiments that produce different results even when carried out under identical conditions.  Indeed, fluctuating phenomena cannot be reproduced at will.  The inability to reproduce the conditions in which an experiment takes place creates the problem of recording the exact time at which the experiment occurred.  One hour is not identical to another.  Simply because phenomena fluctuate.  The date and time characterize a physical situation in space that changes over time.  In chemistry, biology, physics and possibly sociology and psychology, time is a coordinate, not only a duration».

Authors’ Response to Point 4

We understand what the reviewer is saying. The problem is that any kind of criticism we provide must be methodological, or epistemological. Considering the necessary fruitful “contamination” of these approach in hard sciences, is the main focus we are trying to address. However, along section 3 and 4 we deem to have given an extensive description of how the “QFT approach” is supportive of a truly holistic perspective and why the latter must be adopted (unlike any scientific publication in the last decade, at least, is doing). For further details we invite the readers to have a look to references [29][9] and [33][10] as listed in the manuscript. If “demonstrating” requires something like computation about “meaning” and “configuration”, it becomes akin to make a thermometer “measure” the colour of the beech leaves in May, just for example. We are fully aware that our approach may induce the feeling that this is “not science”, but this is not quite our intention or way… On the contrary, we are really trying to rise a methodological (unseen and discarded) problem that has been overlooked and cannot be tackled within the same frame that created it. We really hope that the reviewer may understand the importance of what we’re saying and support us in transmitting this delicate point to other colleagues in order to promote a renewed scientific sensitivity.

Authors’ Response to Point 5

Thank you, we try to shorten the sentences, and to fix the punctuation mistakes, and improve some phrasing. If there are points to highlight where “inconsistent tense usage” is spotted out, please, notify to us and we’ll fix them for sure.

[1] Wheeler JA & Ford Kh (1998). Geons, Black Holes and Quantum Foam. A Life in Physics. WW Norton & Co., New York. ISBN 978-0-393-07948-7. pp. 63-64. & pp. 340-341

[2] Fischer E (1894), Einfluss der Konfiguration auf die Wirkung der Enzyme. Ber. Deut. Chem. Ges., 27: 2985-2983. doi:  10.1002/cber.18940270364

[3] Shannon CE (1948) A Mathematical Theory of Communication. Bell Syst. Techn. J., 27: 379-423; 623-656. doi:  10.1002/j.1538-7305.1948.tb01338.x

[4] Wiener N (1948) Cybernetics or Control and Communication in the Animal and the Machine. MIT Press: Cambridge (MA). doi: 10.7551/mitpress/11810.001.0001

[5] Brillouin L (1951) Maxwell's demon cannot operate: Information and Entropy. J. Appl. Phys., 22:334-337. doi:  10.1063/1.1699951

[6] Fitzroy storm glass https://www.edscuola.it/archivio/lre/on_storm_glass.htm

[7] Manzelli P, Masini G, Costa M (2012) The secrets of water: the scientific work by Giorgio Piccardi. Di Rienzo Publisher, ISBN 888323274-7 (in Italian)

[8] Manzelli P, Masini G, Costa M (1994) I segreti dell’acqua – l’opera scientifica di Giorgio Piccardi (The secrets of water: the scientific work by Giorgio Piccardi. Di Rienzo Publisher, ISBN 88-86044-23-2 (in Italian) …. see p.26, here reported the original text in Italian:

Nel caso dei fenomeni fluttuanti, non potendo fissare le condizioni in cui si svolge un fenomeno, non possono considerarsi valide solo quelle esperienze che danno risultati costanti quando vengono effettuate nelle identiche condizioni tradizionali.  Quindi, come metodo per i fenomeni fluttuanti, dobbiamo ammettere validi gli esperimenti che danno risultati differenti anche quando sono effettuati nelle identiche condizioni tradizionali.  I fenomeni fluttuanti, infatti, non sono riproducibili a piacimento.  Il non poter riprodurre le condizioni in cui si svolge una esperienza ci pone il problema di registrare l’istante ed il periodo di tempo in cui l’esperimento si è svolto.  Un’ora non è identica ad un’altra ora. Proprio perché’ i fenomeni sono fluttuanti.  La data e l’ora caratterizzano una situazione fisica spaziale che cambia nel corso del tempo.  Il tempo in chimica, biologia e fisica e forse anche in sociologia e psicologia, non e soltanto una durata, ma una coordinata.

[9] Madl P, Renati R (2023) Quantum Electrodynamics Coherence and Hormesis: Foundations of Quantum Biology, IJMS 24(18):14003. doi:  10.3390/ijms241814003

[10] Renati P (2022) Relationships and Causation in Living Matter: Reframing Some Methods in Life Sciences?, Phys Scie Biophys J, 6(2): 000217. doi:  10.23880/psbj-16000217

Round 2

Reviewer 1 Report

Comments and Suggestions for Authors

The authors appropriately revised the paper according to the comments, and it is suitable for publicaiton.

Author Response

thank you for your comments and suggestions,

we updated the manuscript including them and more in depth explanations about the structure of the paper and the reasons of its developments. Please, see the attached document for our considerations and replies to your precious comments.
